# Herpesviral lytic gene functions render the viral genome susceptible to novel editing by CRISPR/Cas9

Hyung Suk Oh[1†], Werner M Neuhausser[2,3,4†*], Pierce Eggan[2,3], Magdalena Angelova[1], Rory Kirchner[5], Kevin C Eggan[2,3,6], David M Knipe[1*]

[1]Department of Microbiology, Blavatnik Institute, Harvard Medical School, Boston, United States; [2]Department of Stem Cell and Regenerative Biology, Harvard University, Cambridge, United States; [3]Harvard Stem Cell Institute, Harvard University, Cambridge, United States; [4]Department of Obstetrics and Gynecology, Division of Reproductive Endocrinology and Infertility, Beth Israel Deaconess Medical Center, Harvard Medical School, Boston, United States; [5]Department of Biostatistics, Harvard TH Chan School of Public Health, Boston, United States; [6]Stanley Center for Psychiatric Research, Broad Institute of MIT and Harvard, Cambridge, United States

**Abstract** Herpes simplex virus (HSV) establishes lifelong latent infection and can cause serious human disease, but current antiviral therapies target lytic but not latent infection. We screened for sgRNAs that cleave HSV-1 DNA sequences efficiently in vitro and used these sgRNAs to observe the first editing of quiescent HSV-1 DNA. The sgRNAs targeted lytic replicating viral DNA genomes more efficiently than quiescent genomes, consistent with the open structure of lytic chromatin. Editing of latent genomes caused short indels while editing of replicating genomes produced indels, linear molecules, and large genomic sequence loss around the gRNA target site. The HSV ICP0 protein and viral DNA replication increased the loss of DNA sequences around the gRNA target site. We conclude that HSV, by promoting open chromatin needed for viral gene expression and by inhibiting the DNA damage response, makes the genome vulnerable to a novel form of editing by CRISPR-Cas9 during lytic replication.

**\*For correspondence:**
wneuhaus@bidmc.harvard.edu (WMN);
david_knipe@hms.harvard.edu (DMK)

[†]These authors contributed equally to this work

## Introduction

Herpes simplex virus (HSV) 1 and 2 are prevalent neurotropic DNA viruses that cause significant morbidity and mortality in neonates and adults (*Roizman et al., 2013*). During the initial acute primary infection, which may be asymptomatic, HSV undergoes lytic replication in epithelial tissues followed by spread to sensory neurons, where the virus establishes life-long latent infection. Latent infection may be interrupted periodically by reactivation, which can cause significant morbidity such as corneal blindness secondary to HSV keratitis or recurrent genital herpes. Following primary infection or reactivation, immunosuppressed populations such as neonates and transplant patients can progress to severe manifestations such as encephalitis, which carries a high mortality and a high incidence of significant neurologic sequelae in survivors (*Kimberlin, 2004*). In addition, HSV-2 infection increases the risk of HIV transmission (*Freeman et al., 2006*; *Looker et al., 2017*; *Wald and Link, 2002*). Although an approved vaccine is not available, several anti-HSV drugs can be used to treat lytic infection by targeting viral enzymes expressed during the lytic replication cycle. Because latent HSV genomes express minimal levels of viral lytic genes, the number of therapeutic targets is limited. Currently available agents do not prevent establishment of latency and are ineffective at clearing

latent HSV and preventing reactivation. While several other approaches have been proposed to suppress reactivation, including epigenetic drug-mediated suppression and miRNAs (*Arbuckle et al., 2017*; *Flores et al., 2013*; *Hill et al., 2014*), these methods would require lifelong treatment. Thus, there is a strong need for more effective and specific therapeutic strategies to target the latent or reactivation stages of HSV infection.

Genome editing technologies have evolved rapidly in the past decade and include zinc-finger nucleases (ZFNs), transcription activator-like effector nucleases (TALENs), meganucleases, and clustered regularly-interspaced short palindromic repeats (CRISPR)/Cas9 (*Boch et al., 2009*; *Christian et al., 2010*; *Cong et al., 2013*; *Miller et al., 2007*; *Miller et al., 2011*; *Moscou and Bogdanove, 2009*; *Porteus and Baltimore, 2003*; *Sander et al., 2011*; *Stoddard, 2005*; *Wood et al., 2011*; *Zhang et al., 2011*). The CRISPR/Cas9 system, consisting of a bacterial nuclease - guide RNA (Cas9-gRNA) complex, evolved in archaea and bacteria as a form of adaptive immunity to provide a defense mechanism against bacteriophage infection and plasmid transformation (*Bhaya et al., 2011*; *Doudna and Charpentier, 2014*; *Gaj et al., 2013*; *Hsu et al., 2014*). Subsequently, CRISPR/ Cas9 has been engineered to efficiently induce genome modifications in mammalian cells (*Cong et al., 2013*; *Mali et al., 2013*). Because the specificity of DNA cleavage by CRISPR/Cas9 is encoded in a single gRNA (sgRNA) sequence that can be modified easily, this system has experienced rapid uptake in the scientific community and has become an indispensable tool in molecular biology. Cas9 introduces double-stranded DNA breaks, which triggers cellular DNA repair mechanisms including error-prone non-homologous end-joining (NHEJ) and homology-directed repair (HDR) (*Chapman et al., 2012*; *Symington and Gautier, 2011*; *Wyman and Kanaar, 2006*). During NHEJ, random insertions and deletions (indels) can occur, introducing mutations and frameshifts in the target DNA. The activity of individual sgRNAs in vivo may be affected by differential sgRNA stability as well as the efficacy of Cas9-sgRNA binding to genomic DNA in different cell types. CRISPR/ Cas9 editing requires efficient access to target sites within chromatin, and evidence suggests that chromatin modifications and DNA packaging can block genome editing in eukaryotic cells (*Bultmann et al., 2012*; *Chen et al., 2016*; *Horlbeck et al., 2016*; *Hsu et al., 2013*; *Valton et al., 2012*; *Wu et al., 2014*).

HSV-1 DNA genomes have no histones associated with them in the virion, but upon entry into the cell nucleus, viral DNA rapidly becomes associated with histones (*Cliffe and Knipe, 2008*; *Lee et al., 2016*; *Oh and Fraser, 2008*). Differential host cell mechanisms regulating chromatin assembly on viral DNA in permissive (epithelial) versus non-permissive (neuronal) cells contribute to the lytic versus latent infection decision by HSV (*Knipe and Cliffe, 2008*). During establishment of latent infection, HSV lytic promoters are progressively associated with histones that exhibit modifications indicative of heterochromatin (*Cliffe et al., 2009*; *Wang et al., 2005*). Latent viral chromatin is characterized by a more compacted chromatin structure than that of replicating viral DNA (*Wang et al., 2005*) and thus may show reduced Cas9/sgRNA access to DNA target sites (*Horlbeck et al., 2016*).

Recent studies have demonstrated that CRISPR/Cas9 can impair *active* HSV replication in vitro by targeting specific DNA sequences encoding viral proteins (*Roehm et al., 2016*; *van Diemen et al., 2016*). Because lytic and latent HSV genomes contain different levels of nucleosome loading, we attempted to identify more efficient CRISPR/Cas9 reagents to investigate whether latent HSV genomes can be targeted by Cas9. We designed an in vitro screening strategy for sgRNAs and then for the ability to edit quiescent and lytic HSV genomes in human fibroblasts. Here, we report the identification of specific sgRNAs that edit quiescent HSV-1 genomes and inhibit reactivation of quiescent genomes as well as lytic replication. Our results define differences in the mechanisms of editing of lytic and quiescent HSV-1 genomes, highlighting the vulnerability of the lytic viral genome due to viral proteins reducing loading of histones on viral DNA, inhibiting host DNA repair mechanisms, and promoting viral lytic DNA replication.

## Results

### Identification of sgRNAs targeting HSV-1 genomic sequences in vitro

As the first step in identifying sgRNAs that could efficiently cleave lytic as well as quiescent HSV genomes in human cells, we measured the endonuclease activity of sgRNA candidates in an in vitro cleavage assay. To perform a comprehensive analysis of sgRNAs targeting the 5' region of four HSV-

**Table 1.** CRISPR/Cas9 target sequences.

| Name | Efficiency of cleavage | SaCas9 sgRNA + PAM (g was added as needed) | Target sequences (GenBank: KT899744) |
|---|---|---|---|
| **UL30** | | | |
| UL30-1 | + | GCGTCCCGACTGGGGCGAGGT AGGGGT | 62811–62831 |
| UL30-2 | ++ | gAAGTTTTGCCTCAAACAAGGC GGGGGT | 62779–62799 |
| UL30-3 | - | GCGGCGTGGACCACGCCCCGG CGGGGT | 63060–63080 |
| UL30-4 | ++ | gTGCCCCCCCGGAGAAGCGCG CCGGGGT | 62923–62942 |
| UL30-5 | ++ | gACACGTGAAAGACGGTGACG GTGGGGT | 63097–63116 |
| UL30-6 | + | gACCAGCCGAAGGTGACGAAC CCGGGGT | 63595–63614 |
| UL30-7 | ++ | GGCCATCAAGAAGTACGAGGG TGGGGT | 63532–63552 |
| UL30-24* | ++ | gAAACCCCAAAAGCCGCTTGGG TGGGAT | 62589–62609 |
| UL30-25* | + | gCCACCCGAACCCCTAAAGAGG GGGGAT | 62637–62657 |
| UL30-26* | - | gCATGCCGGCCCGGGCGAGCCT GGGGGT | 62542–62562 |
| UL30-27* | ++ | gCCATCCCACCCAAGCGGCTTT TGGGGT | 62581–62601 |
| **UL29** | | | |
| UL29-1 | ++ | gTCAAGGTCCCCCCCGGGCCCC TGGGAT | 61861–61881 |
| UL29-2 | + | GTGTTTGAGGTCGCCGGGCCG GGGGGT | 61502–61522 |
| UL29-3 | ++ | GCCAGCCAGGGTAAGACCCCG CGGGGT | 61028–61048 |
| UL29-4 | ++ | GCCGCCGTCGCGCCCACCCCG CGGGGT | 61007–61027 |
| UL29-14* | ++ | GAGGGTGGGAGACCGGGGTTG GGGAAT | 62029–62049 |
| UL29-15* | ++ | GTCGGGCGTCCGTCGTCGTGC CGGGAT | 61952–61972 |
| UL29-16* | ++ | gCGGGGGTTGTCTGTGAAGGGT AGGGAT | 62064–62084 |
| UL29-17* | ++ | gATCGGCACCCCGTGGTTACCC GGGGGT | 62084–62104 |
| UL29-18* | ++ | gCAGACAACCCCCGGGTAACCA CGGGGT | 62072–62092 |
| UL29-19* | ++ | GGACCCCGCGTTGCCAGCCGC CGGGGT | 62113–62133 |
| UL29-20* | ++ | GAACCCCGGCGGCTGGCAACG CGGGGT | 62105–62125 |
| UL29- 21 | ++ | GGTTCTCGCACGACGGGGCTC GGGGGT | 61685–61705 |
| **UL54** | | | |
| UL54-1* | ++ | GCTGTCGGCTGCCGTCGGGGC TGGGGT | 113541–113561 |
| UL54-2 | ++ | gACCTGGAATCGGACAGCAAC GGGGAGT | 113667–113687 |
| UL54-3 | ++ | GCTCCGGTCCGTCCTCTCCGT GGGGGT | 113728–113748 |
| UL54-4 | ++ | GCGTCTGGGTGCTGGGTACGC CGGGGT | 113803–113823 |
| UL54-5 | ++ | GGCGGACGCCGTGGGCGTCGC AGGGGT | 113982–114002 |
| UL54-6 | ++ | gTGGTTCTGGGGGCACGCCGGC GGGGGT | 114055–114075 |
| UL54-7 | ++ | GCAGGCTGGGCTTTGGTCGGT GGGGGT | 113957–113977 |
| UL54-8 | ++ | gCGCCGTGGGCGTCGCAGGGGT CGGGGT | 113988–114008 |
| UL54-9 | + | GTCCGTCCACCCCGCCCCGGGG CGGGGT | 114098–114119 |
| UL54-14* | ++ | gCGCTTCCGCGGGGACCCGGGC GGGGGT | 113234–113254 |
| UL54-15* | ++ | gCGCCCGGGGGGCGGAACTAGG AGGGGT | 113347–113367 |
| UL54-16 | ++ | GGCGGCTCTCCGCCGGCTCGG GGGGGT | 113641–113661 |
| **Rs1** | | | |
| Rs1-1 | ++ | GCCGGGCGTCGTCGAGGTCGT GGGGGT | 130775–130795, 146992–147012 |
| Rs1-2 | - | gCCGCTCGTCGCGGTCTGGGCT CGGGGT | 130866–130886, 146901–146921 |
| Rs1-3 | - | GGGGGTGGTCGGGGTCGTGGT CGGGGT | 130796–130816, 146971–146991 |
| Rs1-4 | ++ | gATCGTCGTCGGCTAGAAAGGC GGGGGT | 130599–130619, 147168–147188 |
| Rs1-5 | ++ | GGCGCGGCGACAGGCGGTCCG TGGGGT | 130475–130495, 147292–147312 |

*Table 1 continued on next page*

*Table 1 continued*

| Name | Efficiency of cleavage | SaCas9 sgRNA + PAM (g was added as needed) | Target sequences (GenBank: KT899744) |
|---|---|---|---|
| Rs1-6 | ++ | GCGAGGCCGCGGGGTCGGGCGT CGGGAT | 130634–130655, 147132–147153 |
| Rs1-7 | + | GGGTCCGGGGCGGCGAGGCCG CGGGGT | 130622–130642, 147145–147165 |
| Rs1-8 | ++ | gCGCGAGGCGCGGGCCGTCGGG CGGGGT | 130290–130310, 147477–147497 |
| Rs1-9 | ++ | GCGGACGACGAGGACGAGGACC CGGAGT | 130378–130399, 147388–147409 |
| Rs1-15* | ++ | GCCGATGCGGGGCGATCCTCC GGGGAT | 130954–130974, 146813–146833 |
| Rs1-16* | - | gTACGCGGACGAAGCGCGGGAG GGGGAT | 131142–131162, 146625–146645 |
| Rs1-17* | + | gCGCGTCGACGGCGGGGGTCGT CGGGGT | 131061–131081, 146706–146726 |
| Rs1-18* | ++ | GCGCTAGTTCCGCGTCGACGGC GGGGGT | 131070–131091, 146696–146717 |
| UL26-27 | ++ | GAGGAAATCGGCACTGACCAA GGGGGT | 52742–52762 |
| UL37-38 | ++ | GTATAACACCCCGCGAAGACG CGGGGT | 84066–84086 |

++:full cleavage (no residual substrate DNA on agarose gel), +: partial cleavage (some residual substrate DNA on agarose gel), -: no cleavage, *: non-coding region

1 essential genes, $U_L29$, $U_L30$, $U_L54$/ICP27, $R_S1$/ICP4, we designed a total of 58 sgRNAs (11–13 per gene, *Table 1*) within a 1one kbp span of the 5' coding region of each gene. We reasoned that the in vitro cleavage assay would allow us to measure the influence of sgRNA stability, Cas9-loading and target sequence composition independently of transcription/turnover rates and access to target sites in genomic viral DNA within cells.

To enhance the potential for in vivo delivery of CRISPR-Cas9 in future experiments, we designed sgRNAs compatible with *Staphylococcus aureus* Cas9 (SaCas9). SaCas9 is small enough to be encoded by adeno-associated virus (AAV)-based delivery systems and has been used to transduce external genes in mouse brain in vivo (*Ran et al., 2015*). Because SaCas9 protein was not commercially available when we started this study, we selected SaCas9 sgRNAs with protospacer adjacent motif (PAM) sites that are compatible with *Streptococcus pyogenes* Cas9 (SpCas9). PAM sequences of SpCas9 (NGG) and SaCas9 (NNGRRT/N) are not mutually exclusive (*Ran et al., 2013*; *Xie et al., 2018*), and cleavage efficiencies of SpCas9 and SaCas9 are comparable at the sites of shared PAM sequences (*Friedland et al., 2015*). To measure sgRNA activity, we incubated individual sgRNAs/SpCas9 protein complexes together with PCR-generated DNA substrates containing the sgRNA target viral sequence and measured DNA cleavage by gel electrophoresis relative to controls (*Figure 1A*). We found that certain sgRNAs promoted cleavage more efficiently than others (*Figure 1B* and *Table 1*). We then chose a set of sgRNAs listed below that performed best in the in vitro cleavage assay for further analysis in human cell-based systems.

## CRISPR/Cas9 inhibits HSV lytic infection

To evaluate the effects of our set of in vitro screened sgRNAs on HSV-1 lytic replication, we transduced human foreskin fibroblasts (HFFs) with lentiviruses expressing SaCas9 and sgRNAs in the presence of puromycin for 7 days (d), infected with WT HSV-1 at a multiplicity of infection (MOI) of 0.1 or 5, and harvested the infected cells at 48 hrpost infection (hpi) or 24 hpi, respectively (*Figure 2A*). All of the sgRNAs (with the exception of *UL29-4)* reduced viral yields by 2–4 logs at low MOI (0.1, *Figure 2B*) and at high MOI (5, *Figure 2C*). *UL30-5* was the most efficient in reducing viral titers by more than three logs at an MOI of 5 and more than four logs at an MOI of 0.1. We then evaluated dual sgRNAs targeting two different genes and found additive reductions in viral yields with the effect being less pronounced at high MOI than at low MOI (*Figure 2B and C*). These results demonstrated that our sgRNAs targeting HSV-1 genes can inhibit lytic replication in human cells, consistent with previous reports (*Roehm et al., 2016*; *van Diemen et al., 2016*). In addition, simultaneous targeting of multiple essential HSV genes significantly improves the efficiency of viral inhibition compared to targeting of single genes.

To examine the specificity of CRISPR/Cas9 editing on viral protein expression, we infected the SaCas9/sgRNA-expressing cells with WT HSV-1 at an MOI of 5, harvested protein lysates at 10 hpi,

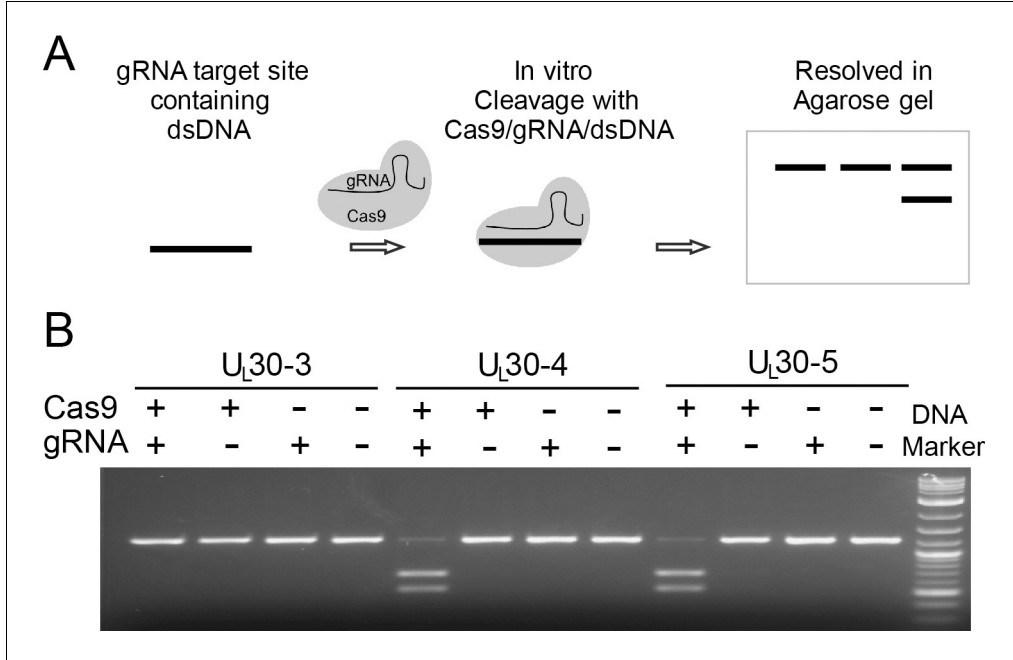

**Figure 1.** In vitro cleavage assay. (A) Schematic diagram of in vitro cleavage assay (B) Results are shown for three sgRNAs targeting $U_L30$ (UL30-3, -4, and -5). T7 in vitro transcribed sgRNA was combined with SpCas9 protein and a PCR template containing the CRISPR sequence, incubated 1 hr at 37°C and run on an agarose gel. Lane (1) SpCas9+sgRNA, lane (2) Cas9 only, lane (3) sgRNA only, lane (4) no Cas9/sgRNA. Efficient cutting is seen for UL30-4 and -5 but not UL30-3.

and measured HSV-1 protein levels by immunoblotting. The sgRNAs targeting the immediate-early (IE) $R_S1$ and $U_L54$ genes reduced levels of the corresponding proteins ICP4 and ICP27, compared to the protein levels of SaCas9 expressing control cells (*Figure 2D*). The sgRNAs, except for UL29-4, targeting early (E) genes $U_L29$ and $U_L30$ reduced levels of the corresponding proteins, ICP8 and UL30. Interestingly, the sgRNAs targeting $U_L30$ also reduced levels of ICP8 significantly and reduced ICP4 and ICP27 slightly. These results confirmed that the selected sgRNAs can reduce the expression levels of the proteins encoded by the targeted genes. Although the in vitro cleavage assay showed equivalent cleavage efficiencies of the selected sgRNAs (*Table 1*), we observed that two sgRNAs targeting the same viral gene, for example UL29-3 and UL29-4, had different efficiencies of reduction of ICP8 protein levels, which we also observed in the indel mutation rates of UL29-3 and UL29-4 in quiescent *d*109 infected HFFs (see below). It is possible that different expression levels of SaCas9 with different sgRNAs affected the cleavage efficiency of CRISPR/Cas9. To examine this possibility, we measured SaCas9-HA levels by immunoblotting using an anti-HA antibody. The levels of SaCas9-HA varied slightly in different SaCas9/sgRNA expressing cells but did not correlate with the level of the targeted protein (*Figure 2D*). The lowest level of SaCas9-HA in UL54-3 expressing cells showed equivalent reduction of ICP27 compared to UL54-2 expressing cells, arguing that the level of SaCas9 with different sgRNAs was not the major contributor to the difference of knockout efficiency in these experiments.

We also transduced HFFs with combinations of sgRNAs to evaluate the effect of dual sgRNAs. We transduced UL30-5 in combination with each of the other sgRNAs tested above and found that all of the sgRNAs reduced the levels of target gene proteins, except for UL29-4, as expected (*Figure 2D*). These results indicated that expression of specific HSV proteins can be reduced by targeting the viral genes encoding the corresponding proteins using sequence-specific single or double sgRNAs.

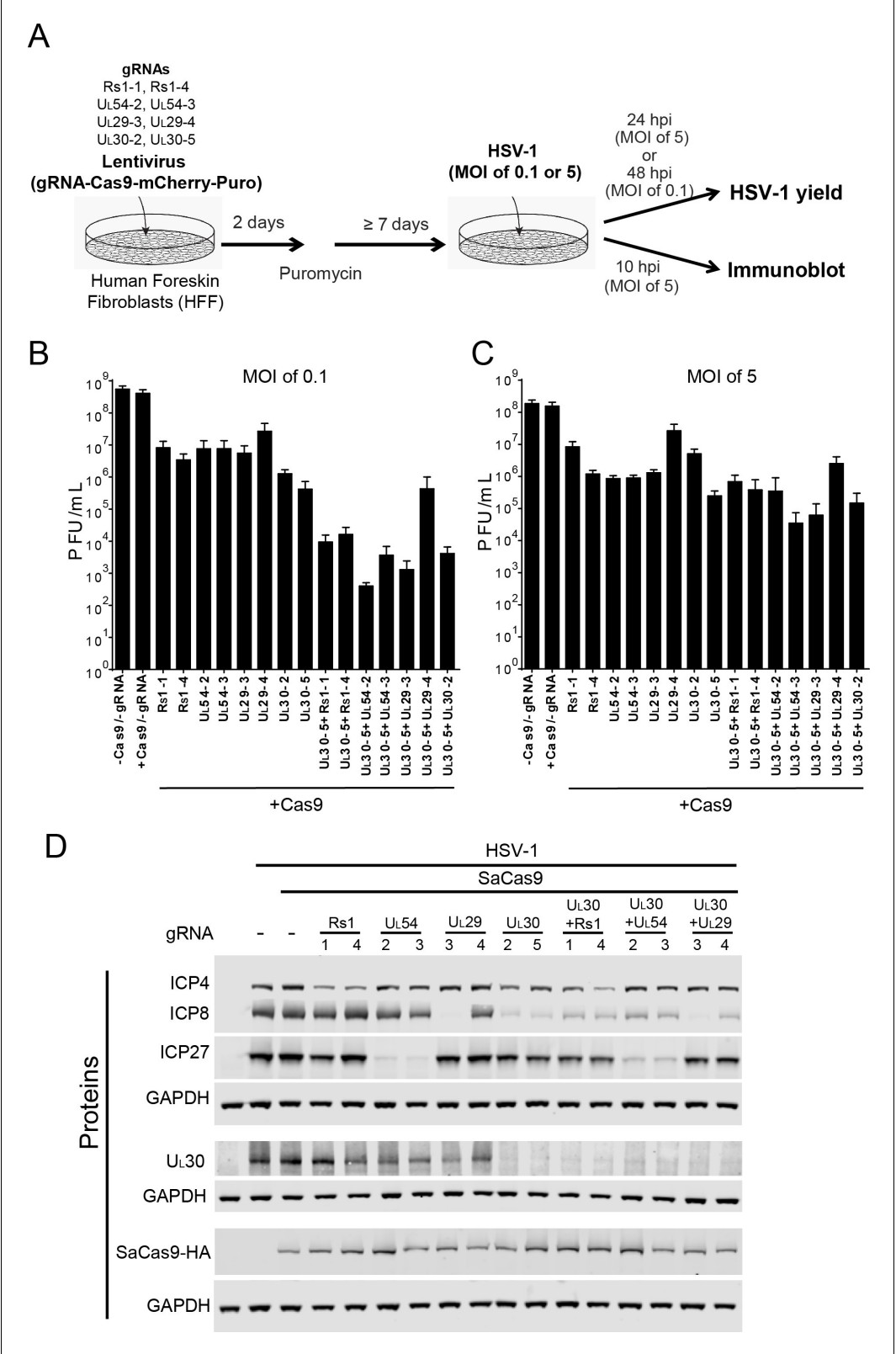

**Figure 2.** Effect of CRISPR-Cas9 on HSV-1 lytic infection. (**A**) Experimental scheme of SaCas9/sgRNA-mediated inhibition of HSV lytic infection. (**B and C**) HFFs transduced with lentivirus expressing SaCas9 and sgRNAs were infected with HSV-1 at an MOI of 0.1 (**C**) or 5 (**D**) and harvested at 48 hpi or 24 hpi, respectively. Viral yields were determined by plaque assays. The histogram shows the mean values and standard deviations of biological replicates at an MOI of 0.1 (N = 3) or at an MOI of 5 (N = 4). All the sgRNA added conditions showed statistical significance compared to +Cas9 /-gRNA (one-way

*Figure 2 continued on next page*

*Figure 2 continued*

ANOVA with Dunnett's multiple comparisons test, p<0.0001 (MOI of 0.1) and p<0.01 (MOI of 5, except for UL29-4 (p<0.05))). (D) HFFs transduced with lentivirus expressing SaCas9 and sgRNA were infected with HSV-1 at an MOI of 5 and harvested at 10 hpi. Proteins were detected using immunoblotting with antibodies specific for the indicated proteins. Three SDS-PAGE gels loaded with the same amount of proteins were used to detect multiple proteins. Immunoblots of GAPDH are shown as a control under the individual immunoblots.

## CRISPR/Cas9 inhibits reactivation of quiescent HSV-1 genomes

To study the effect of our set of sgRNAs on latent HSV-1, we employed a quiescent infection system with replication-defective HSV-1 *d*109 virus, which shows heterochromatin loading on viral DNA (*Ferenczy and DeLuca, 2009*) similar to murine latent infection (*Cliffe et al., 2009*; *Wang et al., 2005*). We infected HFFs with HSV-1 *d*109 virus at an MOI of 10 and allowed quiescent infection to be established for 7–10 d. The cells were then transduced with different lentiviruses expressing SaCas9 and individual sgRNAs that had performed well in the in vitro cleavage screen for HSV-1 E genes $U_L29$ and $U_L30$. To induce reactivation, we super-infected quiescently infected HFFs with

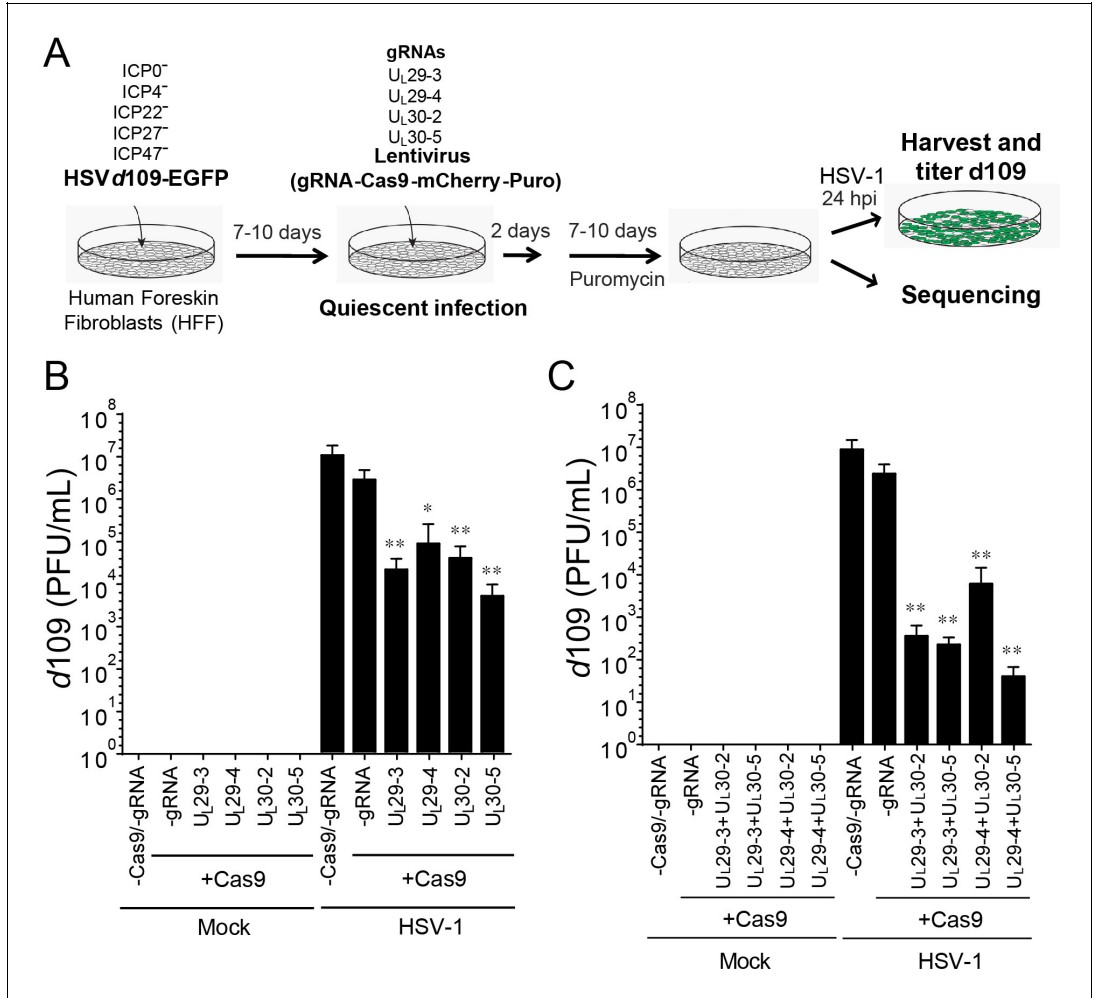

**Figure 3.** CRISPR/Cas9-induced mutagenesis of quiescent *d*109 genomes and effect on reactivation. (**A**) Experimental scheme of SaCas9/sgRNA-mediated inhibition of reactivation of quiescent *d*109 genomes in HFFs. HFFs were infected with HSV-1 *d*109 virus to establish quiescent infection for 7–10 d and transduced with lentivirus expressing SaCas9 and sgRNAs for 7–10 d. (**B and C**) HFF were infected with *d*109 to establish quiescent infection for 7–10 d and transduced with lentivirus expressing SaCas9 and sgRNAs for 7–10 d. To reactivate quiescent *d*109 genomes, HFFs were superinfected with WT HSV-1 at an MOI of 5 and harvested at 24 hpi. GFP-positive viral yields were determined by plaque assays on FO6 and V27 cells. The histogram shows the mean values and standard deviations of biological replicates (B and C: N = 5 and N = 7 respectively). All the sgRNA added conditions showed statistical significance compared to +Cas9 /-gRNA (Ratio paired t test, *p<0.05 and **p<0.01).

wildtype (WT) HSV-1 (*Figure 3A*). In this set of experiments, *d*109 viral reactivation was quantified using plaque assays by counting GFP-positive plaques on complementing FO6 cells. FO6 is a Vero-derived cell line expressing ICP4, ICP27, and ICP0 upon HSV-1 infection, thereby complementing replication of *d*109 virus (*Samaniego et al., 1998*). However, recombination between WT HSV-1 and *d*109 could transfer the GFP sequence to WT HSV-1 and result in production of false GFP-positive plaques. To measure these recombinant viruses, we also counted GFP-positive plaques formed on V27 cells, which express ICP27 (but not ICP4 and ICP0) upon infection with HSV (*Rice and Knipe, 1990*). Because *ICP27* is replaced with *GFP* in HSV-1 *d*109, any recombinant GFP-positive but ICP27-negative HSV mutants that arise (but not HSV-1 *d*109), can replicate in V27 cells. To calculate the number of plaques originating from reactivated *d*109 genomes, we subtracted the number of GFP-positive plaques on V27 cells from the number of GFP-positive plaques on FO6 cells.

Mock-infected HFFs containing quiescent HSV-1 showed no signs of HSV-1 whereas superinfection of these cells with WT HSV-1 resulted in robust viral reactivation of *d*109 (*Figure 3B*). However, transduction with anti-HSV-1 sgRNAs UL30-2, UL30-5, UL29-3 or UL29-4 caused a more than 100-fold reduction in reactivation of quiescent *d*109 virus (*Figure 3B*). To evaluate the potential additive effect of the sgRNAs, we transduced quiescent *d*109-infected HFF cells with two lentiviruses expressing SaCas9/sgRNAs targeting the $U_L29$ and $U_L30$ genes, and reactivated the quiescent *d*109 genomes as described above. Consistent with the independent roles of the pairs of gene products, we observed that the cells expressing two different sgRNAs showed greater reduction (over 10-fold additive reduction compared to single sgRNA-transduced cells) in reactivation of quiescent *d*109 genomes (*Figure 3C*). Interestingly, in more than one half of the experiments, we found that cells transduced with two lentiviruses expressing UL29-4 and UL30-5 yielded very low numbers of GFP-positive plaques on FO6 or V27 cells after superinfection with WT HSV-1. This suggested a near-complete inhibition of reactivation of quiescent *d*109 genomes (*Figure 3C*). These results demonstrated that anti-HSV-1 sgRNAs can reduce reactivation of quiescent *d*109 genomes, with combinations of two different sgRNAs exhibiting an additive effect.

## CRISPR/Cas9 induces indel mutations in quiescent HSV-1 genomes

A previous study demonstrated that CRISPR/Cas9 can reduce the reactivation of HSVs; however, the quiescent viral genomes showed little or no indel formation in these experiments (*van Diemen et al., 2016*). To assess whether CRISPR/Cas9-induced editing of viral genomes occurred at the corresponding CRISPR target sites in our model of quiescent HSV infection, we deep-sequenced PCR-amplicons of target sites from *d*109 virus-infected cells transduced with vectors expressing SaCas9 and HSV-1 sgRNAs or just SaCas9. No indels were detected in cells expressing SaCas9 only (Control; *Figure 4A*). The UL30-5 sgRNA target site showed the highest indel mutation frequency (mean = 59.3 ± 26.85% SD) compared to the other sgRNA target sites with detectable editing frequencies (*Figure 4A*, mean = 53.88% (± 23.62%), 11.05% (± 11.52%) and 5.6% (± 5.56%) for UL29-3, UL29-4, and UL30-2, respectively). The indel analysis and sequence alignment (*Figure 4B and C*) showed that more than 95% of sequence changes induced by SaCas9/UL30-5 sgRNA were less than six nucleotide-indels at the target site. In addition, more than 84% of the indel mutations induced frame shifts resulting in an early termination of $U_L30$ protein expression and the rest of the other mutations introduced insertion or deletion mutations into the $U_L30$ gene.

These results indicated that (1) individual sgRNAs that cleave quiescent HSV genomes efficiently can be identified and (2) SaCas9/sgRNA-induced indel mutations lead to loss-of-function mutations in the corresponding protein.

## CRISPR/Cas9 induces indel mutations in HSV-1 genomes during lytic replication

To define the mechanisms of CRISPR-Cas9 editing of lytic genomes, we analyzed the effect of SaCas9/sgRNAs on lytic viral genomes by measuring indel rates and levels of HSV-1 genomic DNA sequences during lytic infection with or without viral DNA replication.

First, to define the time course of editing or indel introduction within 24 hr post treatment with SaCas9/UL30-5, we performed deep sequencing of the CRISPR target site from SaCas9/UL30-5-transduced and WT HSV-1-infected HFFs in the presence or absence of the viral DNA synthesis inhibitor, phosphonoacetic acid (PAA), at different time points post infection. We detected indel

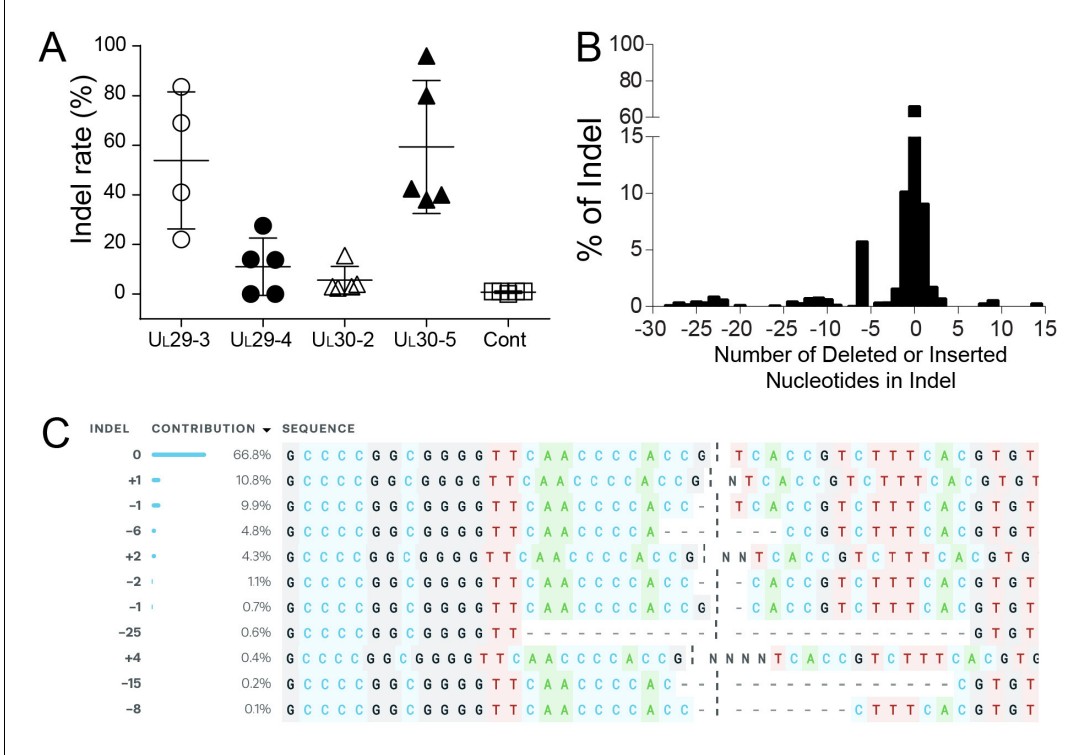

**Figure 4.** Indel mutations in the HSV-1 genome during the quiescent infection. (A) Indel mutation frequencies of quiescent *d*109 genomes are shown at the indicated sgRNA target sites. (B) Histogram representing the frequency (count) of indel lengths induced by SaCas9/UL30-5 in quiescent *d*109 genomes. (<0) deletions (>0) insertions. (C) Examples of sequences that show mutations induced by SaCas9/UL30-5 in quiescent *d*109 genomes.

mutations by four hpi and the levels of indels increased to more than 80% by 24 hpi during lytic replication (*Figure 5A*), equal to or better than the effect of UL30-5 on quiescent genomes (*Figure 4A*). However, in the absence of viral DNA replication, we detected significantly lower levels of indels over a 24 hpi period using deep sequencing (*Figure 5A*, +PAA). Thus, editing was most efficient on replicating or replicated viral DNA.

Second, to characterize the editing of the UL30-5 sgRNA target sequences during lytic replication, we analyzed the indel length at the target sites in viral DNA using PCR-seq. At six hpi, we observed that 36.5% of the sequences of MiSeq results contained an indel mutation, and 99% of the indel mutations were equal to or less than six nucleotides (≤6 NT) in length (*Figure 5A*; *Figure 5B* and *Figure 5—figure supplement 1*). The fraction of the molecules with indels increased to 80% by 12 hpi, and nearly all indels were equal to or less than six nucleotides (≤6 NT) in length (*Figure 5A*; *Figure 5B*).

Third, to test for the loss of sequences at the target site, we used qPCR to measure the levels of viral DNA sequences across the UL30-5 target site (UL30-5 PCR), as well as a non-targeted region within the $U_L29$ gene ($U_L29$ PCR) and with SaCas9 ± UL30-5. We infected SaCas9/sgRNA-transduced cells with WT HSV-1 and harvested total DNA for qPCR at the indicated times post infection. In the absence of sgRNA, we observed that levels of $U_L29$ and $U_L30$ DNA sequences in SaCas9 expressing control cells increased by more than $10^3$-fold by 12 hpi (*Figure 5C*). In contrast, SaCas9/UL30-5-expressing cells showed an increase in the abundance of the $U_L29$ PCR product of less than $10^2$-fold (*Figure 5C*), indicating a 10-fold reduction in HSV genomes compared to control cells. Moreover, in SaCas9/UL30-5 expressing cells, we detected a 100-fold reduction of UL30-5 PCR compared to controls (*Figure 5D*). The reduced levels of $U_L29$ sequences indicated that viral DNA replication was reduced by CRISPR/Cas9, while the greater reduction of $U_L30$ sequences was consistent with loss of sequences across the gRNA target site and/or cleavage and lack of re-ligation.

Fourth, because PCR amplification across the UL30-5 target site cannot distinguish between extensive chew back of cleaved DNAs and non-re-ligated cleaved viral DNAs, we performed an

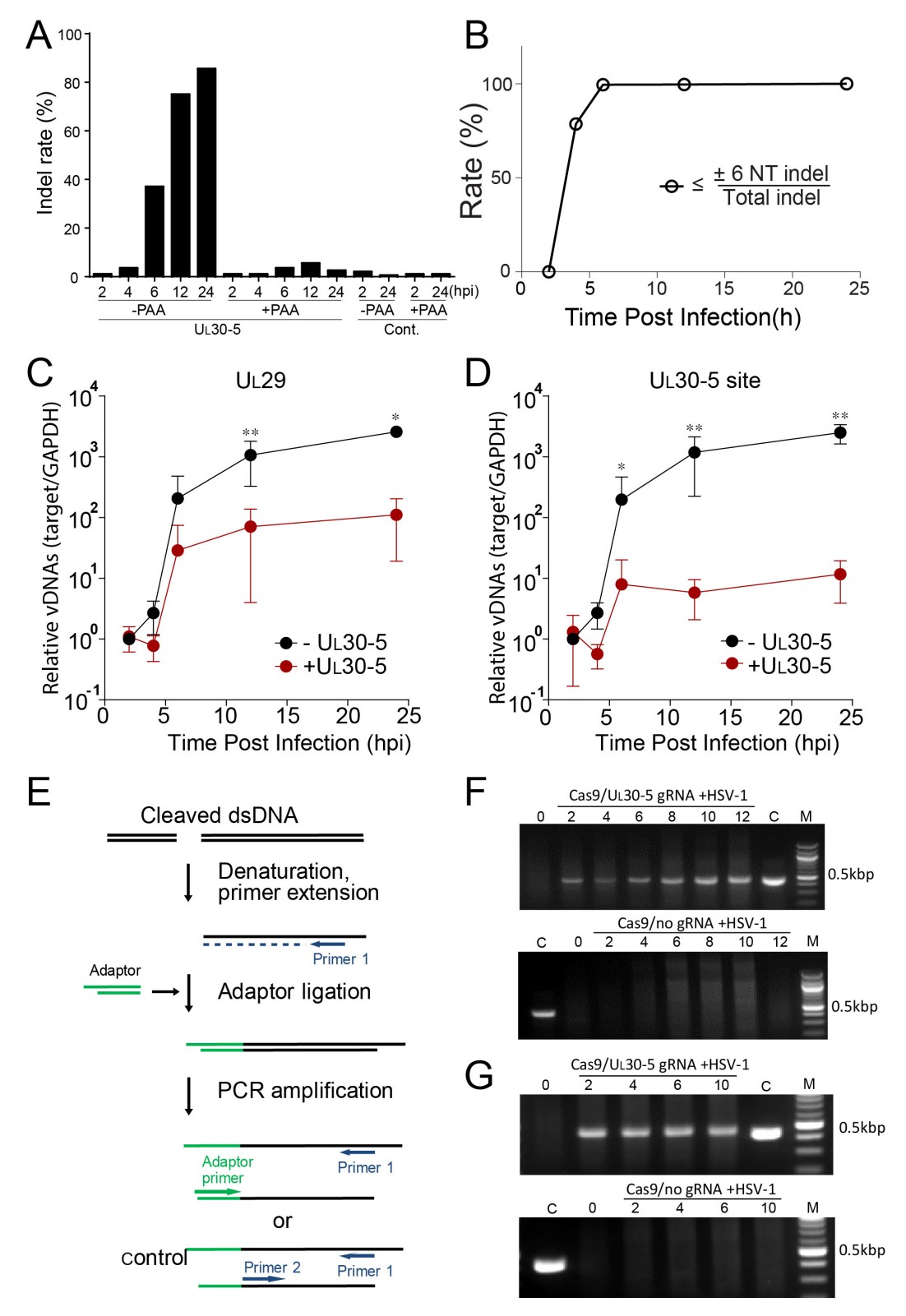

**Figure 5.** Effect of CRISPR-Cas9 on input and replicating HSV genomes. (**A**) Kinetics of indel mutations in the HSV-1 genome during lytic infection. HFFs transduced with lentivirus expressing SaCas9 and sgRNA were infected with HSV-1 at an MOI of 3 in the presence or absence of phosphonoacetic acid (PAA) and harvested at the indicated times post infection. Indel mutation frequencies are shown at the indicated times post infection at the sgRNA target sites by MiSeq. (**B**) Calculated portion of short indels (≤±6 nucleotide (NT) indel) out of total indel during the lytic replication. (**C and D**) HFFs

*Figure 5 continued on next page*

*Figure 5 continued*

transduced with lentivirus expressing SaCas9 and sgRNA were infected with HSV-1 at an MOI of 3 and harvested at the indicated times post infection. The accumulated DNAs were detected by real time qPCR amplification within the $U_L29$ gene (C) or over the UL30-5 sgRNA (D) target site. The histogram shows the mean values and standard deviations of biological replicates (N ≥ 3, Ratio paired t test, *p<0.05 and **p<0.01). Ligation-Mediated -PCR (LM-PCR) of HSV-1 viral DNA during lytic replication. (E) Schematic diagram of LM-PCR (adapted and modified from *Brinkman et al., 2018*). To make blunt end dsDNA, primer extension was performed using cleaved dsDNA and a primer (primer 1) about 500 bp away from the sgRNA target site. An annealed adaptor was ligated to the blunt end dsDNA and PCR amplification was performed using the pair of either the primer 1/adaptor primer or the primer 1/primer 2 for control. (F and G) HFFs transduced with lentivirus expressing SaCas9 and UL30-5 sgRNA were infected with HSV-1 (MOI of 3) in the absence (F) or presence (G) of PAA and harvested at the indicated times post infection. Total DNA was purified, and a primer extension reaction was performed using a complementing primer downstream of the UL30-5 site to convert all the cleaved DNA into blunt end dsDNA. An adaptor was ligated to the blunt end of dsDNA, and PCR was performed using a primer complementing adaptor and the primer used for the extension reaction. Top panel: +UL30-5 sgRNA, bottom: control. C: control PCR product generated using the primer for the extension reaction and a primer complementing near the UL30-5 target site. The right lane (M): DNA ladder.

The online version of this article includes the following figure supplement(s) for figure 5:

**Figure supplement 1.** Histogram of indel length of HSV-1 and short indel accumulation during lytic replication.

assay to detect free DNA ends. We infected SaCas9/UL30-5 sgRNA-expressing cells with HSV-1, isolated DNAs at the indicated time post infection, and assayed for linear viral DNA by ligating a linker onto free DNA ends and performing PCR using primers 430 bp away from the UL30-5 sgRNA target site (*Figure 5E* and *Brinkman et al., 2018*). We observed that in the presence of SaCas9 and the UL30-5 sgRNA, a PCR product appeared by two hpi and increased to eight hpi (*Figure 5F*), indicating an accumulation of a linear DNA species during lytic replication. This was not observed without the sgRNA (*Figure 5F*), showing that this was likely a cleavage product of SaCas9 that was not rejoined. To evaluate whether input viral genome also can be cleaved and remained as linear DNA, we ligated a linker onto the DNAs isolated from cells treated with PAA during the lytic infection. A PCR product appeared by two hpi (*Figure 5G*), which is similar to the result of DNAs from lytic replication (*Figure 5F*). Interestingly, the intensities of PCR products at 2–10 hpi did not change indicating that the majority of input viral genomes was cleaved by SaCas9/sgRNA as early as two hpi. These results supported the idea that a significant fraction of replicating and input viral genomes are not re-ligated after cleavage by CRISPR/Cas9.

Therefore, lytic viral genomes after Cas9 editing were a mixed population of small indels, linearized molecules, and molecules with large deletions of sequences.

## Editing of input viral DNA

Because we had detected only low levels of indels in the presence of PAA, we hypothesized that non-replicating (input) viral genomes might not be targeted efficiently by SaCas9/sgRNA. To evaluate the effect on the input viral genome, we measured viral protein expression with or without viral DNA synthesis. We infected SaCas9/sgRNA-expressing cells with HSV-1 at an MOI of 1 in the presence or absence of the viral DNA polymerase inhibitor PAA, harvested protein lysates at 10 hpi and measured viral protein levels by immunoblotting (*Figure 6A*). Interestingly, the cells expressing sgRNAs targeting the $R_S1$ and $U_L30$ genes showed similar reductions in the corresponding ICP4 and $U_L30$ protein levels in the presence or absence of PAA compared to SaCas9-expressing control cells. These results argued that while replicating virus is much more susceptible to introduction of indels, CRISPR/Cas9 can reduce target protein expression even from input genomes. These results also argued that the majority of input viral genomes is cleaved early post infection (*Figure 5G*). Therefore, the effect of CRISPR/Cas9 appeared to be more than just reducing viral protein expression.

To evaluate the effect of viral replication on the observed reduction of $U_L30$ sequences, we performed PCR amplification across the sgRNA target sites under conditions where viral DNA synthesis was blocked. In the absence of viral DNA replication, with or without UL30-5, infected cells showed a similar decrease in $U_L29$ PCR over 12 hr (*Figure 6B*). This gradual loss of input viral DNA in the absence of viral DNA synthesis has been observed previously (*Gao and Knipe, 1993*). UL30-5 sequences in the input genomes also decreased by 40% without the sgRNA, but interestingly, in the presence of the UL30-5 sgRNA, the UL30-5 PCR product decreased further by 60% (*Figure 6C*). These results supported the idea that the majority of SaCas9/sgRNA-induced cleaved input and

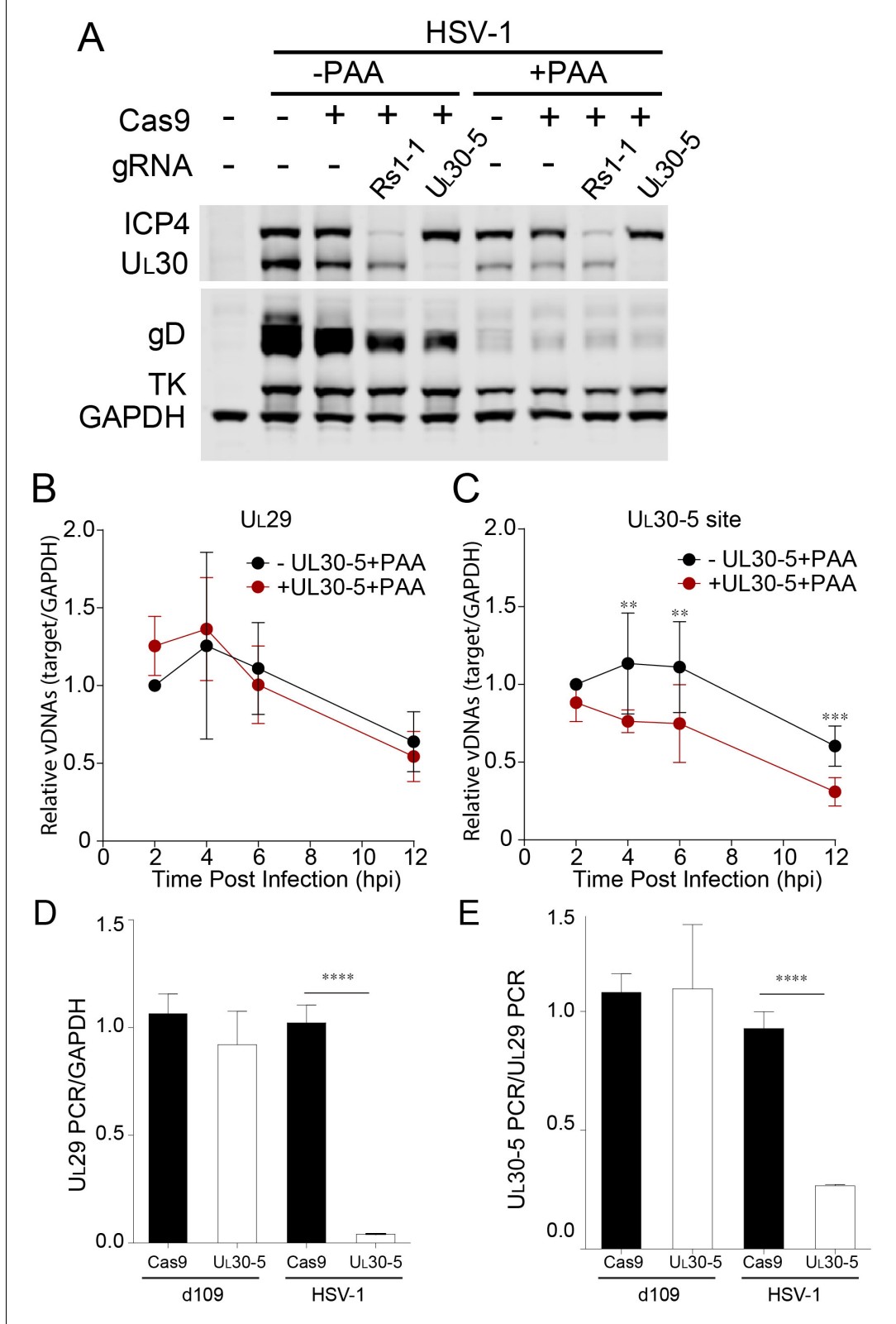

**Figure 6.** Effect of CRISPR-Cas9 on input HSV genomes. (**A**) HFFs transduced with lentivirus expressing Cas9 and sgRNA were infected with HSV-1 at an MOI of 1 in the presence or absence of PAA and harvested at 10 hpi. Proteins were detected by immunoblotting with antibodies specific for the indicated proteins. Immunoblots of GAPDH are shown as a control. (**B and C**) HFFs transduced with lentivirus expressing SaCas9 and sgRNA were infected with HSV-1 at an MOI of 3 in the presence of PAA and harvested at the indicated times post infection. The accumulated DNAs were detected

*Figure 6 continued on next page*

*Figure 6 continued*
by real time qPCR amplifying within the $U_L29$ gene (B) or over the UL30-5 sgRNA (C) target site. The histogram shows the mean values and standard deviations from biological replicates (N ≥ 3, Ratio paired t test, \*\*p<0.01 and \*\*\*p<0.001). (D and E) PCR amplification across the UL30-5 target site in quiescent *d*109 genomes and replicating HSV-1 genomes. Quiescently infected HFF cells were transduced with lentivirus expressing Cas9/UL30-5 sgRNA as described in *Figure 2A* and analyzed at the UL30-5 sgRNA target site by qPCR. The qPCR across the Cas9/UL30-5 sgRNA target site (UL30-5 PCR) was normalized to GAPDH (D) or a remote site of qPCR (E, UL29 PCR) with (UL30-5) or without (Cas9) UL30-5 sgRNA expression. As a control, Cas9 ± UL30-5 sgRNA transduced HFF cells were infected with HSV-1, harvested at 12 hpi and analyzed as described above. Cas9: no sgRNA, UL30-5: Cas9 with UL30-5 sgRNA. The histogram shows the mean values and standard deviations from biological replicates of *d*109 (N = 6) or HSV-1 (N = 3) infected cells (t-test: \*\*\*\*p<0.0001).

replicating (or replicated) viral DNA is not rejoined and/or there is extensive chew-back of the cleaved sequences, either of which would impair the PCR amplification across the UL30-5 sgRNA target site.

Next, we examined whether we could observe a similar reduction of sequences at the UL30-5 sgRNA target site in the SaCas9/sgRNA-treated quiescent genome indicating failure to re-ligate after cleavage. Interestingly, PCR amplification across the UL29 (*Figure 6D*) and UL30-5 (*Figure 6E*) sgRNA target sites did not show a significant difference with or without UL30-5 sgRNA expression. These results supported the idea that cleaved quiescent *d*109 genomes are re-joined without inducing significant loss of sequences or viral DNA degradation during the quiescent infection.

## Editing of lytic HSV genomes causes extensive loss of viral sequences at the target sites

To measure the extent of sequence loss in lytic HSV genomes induced by CRISPR/Cas9 cleavage, we performed whole genome sequencing (WGS) of HFFs infected with WT HSV-1 expressing either SaCas9/UL30-5, SaCas9 only, or a mock transduction (*Figure 7*). We then analyzed the number of WGS reads mapping to the HSV-1 genome. Consistent with our functional analysis (*Figures 2B–D,5C–D,6A*), reads aligning with HSV-1 sequence were strongly reduced in HFFs expressing SaCas9/UL30-5 (4,798,995 reads), compared to SaCas9 only (26,271,968 reads) or a mock trasnduction (47,295,539 reads). Analysis of sequencing coverage confirmed the reduced depth at the UL30-5 cleavage site (*Figure 7A B*). Interestingly, the area of reduced coverage extends to approximately 50,000 bases up-and downstream of the UL30-5 site (*Figure 7A*), indicating large scale (approximately ⅓ of the HSV genome) loss of viral genomic sequence around the UL30-5 cleavage site possibly through degradation.

## sgRNAs targeting non-coding regions of HSV-1 genome reduce viral replication

To dissect the contribution of genomic disruption versus mutagenesis of essential viral genes to the inhibition of HSV replication due to editing, we targeted non-coding sites of the HSV-1 genome using CRISPR/Cas9. We designed two sgRNAs targeting junction sites between the $U_L26$-$U_L27$ or the $U_L37$-$U_L38$ genes, sites which are not essential for viral replication (*Balliet et al., 2007*; *Lin et al., 2016*). We infected SaCas9/sgRNA-transduced cells with WT HSV-1 at an MOI of 0.1 or five and harvested cells at 48 hpi or 24 hpi, respectively. UL30-5 reduced viral yields by more than 3–4 logs as observed previously (*Figure 8A B*). Interestingly, the two sgRNAs targeting intergenic sites, UL26-27 and UL37-38, also showed a significant (10-fold) reduction in viral yields indicating that targeting within nonessential sequences can reduce viral replication. To test the possibility that UL26-27 and UL37-38 sgRNAs affected viral gene expression, we analyzed viral protein expression by immunoblotting. (*Figure 8C*). As expected, cells expressing sgRNAs targeting $U_L30$ showed reduction of the UL30 protein. However, UL26-27 and UL37-38 sgRNAs did not result in any significant changes in ICP4, ICP27, ICP8, or UL30 protein levels. We then tested how the UL26-UL27 and UL37-UL38 sgRNAs affected viral DNA levels. We infected SaCas9/sgRNA transduced HFFs with WT HSV-1 and quantified viral DNA sequences by qPCR. UL30-5 expressing cells showed significantly less PCR amplification with non-target sequences (*Figure 8D*), but UL26-27 and UL37-38 expressing cells showed similar levels of PCR amplification compared to control (-sgRNA) at 24 hpi. Consistent with our previous results, UL30-5 expressing cells showed only a 10-fold increase in $U_L30$ sequences, but in the cells with UL26-27 or UL37-38 sgRNAs, $U_L30$ sequences increased by $4 \times 10^2$ fold, or $3 \times 10^2$

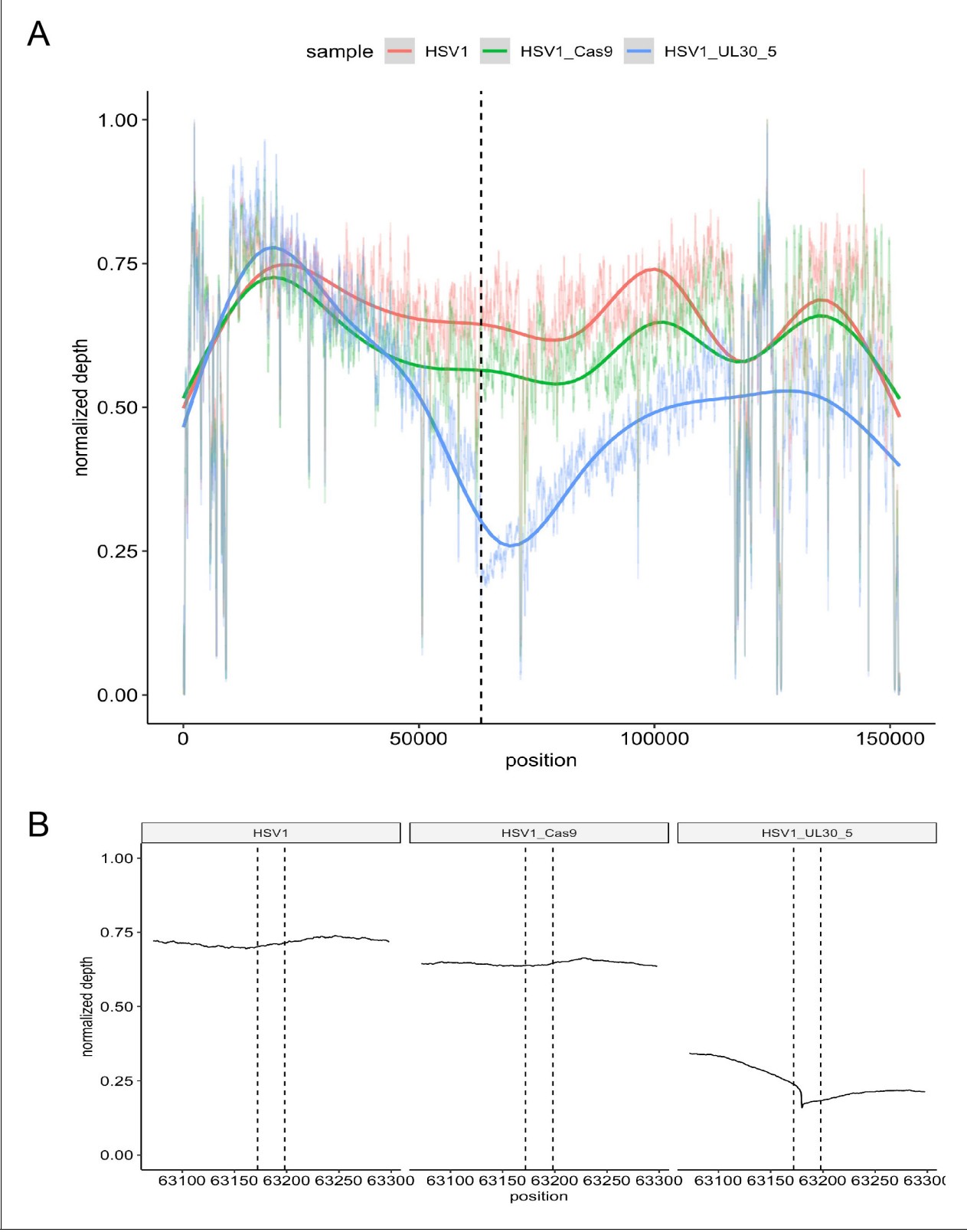

**Figure 7.** On-target activity of UL30-5 sgRNA within the HSV genome sequence during lytic replication. Per-base plot of WGS coverage over each specific base in HSV during lytic replication in HFFs for UL30-5 against untreated controls (no sgRNA, no SaCas9/sgRNA). (**A**) Per-base WGS coverage across the entire HSV genome with fitted LOESS curves obtained by local regression of per-base data (**B**) Zoomed in view of the WGS coverage in the

*Figure 7 continued on next page*

*Figure 7 continued*

genomic area around the UL30-5 cleavage site within $U_L30$ for UL30-5 treated against untreated controls. The vertical lines demarcate the UL30-5 target sequence in each sample.

The online version of this article includes the following source data for figure 7:

**Source data 1.** List of 437 possible UL30-5 off-target sites within the human genome (GRCh38/hg38).

fold respectively (*Figure 8E*), which is similar to $U_L29$ sequences (*Figure 8D*). Similarly, targeting HSV-1 with UL26-27 and UL37-38 resulted in significantly less PCR amplification across their respective target sequences compared to non-targeted regions (*Figure 8F G* compared to *Figure 8D*). These results suggested that CRISPR/Cas9 targeting of non-coding sites in the HSV-1 genome reduces viral replication by disruption of the viral DNA and/or deletion of neighboring genes.

Because quiescent *d*109 genomes targeted by Cas9/sgRNA are re-ligated, we hypothesized that targeting non-coding sites of the HSV-1 genome using CRISPR/Cas9 may not reduce reactivation of quiescent viral genomes. To test this hypothesis, we first examined the level of SaCas9/UL26-27 or SaCas9/UL37-38 sgRNA targeted quiescent *d*109 genomes. As we observed previously with UL30-5 sgRNA (*Figure 6D*), PCR amplification across the UL26-27 or UL37-38 sgRNA target sites did not show a significant difference with or without corresponding sgRNA expression, again indicating re-ligation of quiescent genomes (*Figure 8—figure supplement 1A*). However, reactivation induced by wildtype HSV-1 superinfection showed significant reduction with UL26-27 and UL37-38 sgRNA (*Figure 8—figure supplement 1B*) similar to lytic replication (*Figure 8D*). Because cleavage by SaCas9 destroys the sgRNA target site, these results suggested that HSV genomes escaping cleavage in the quiescent phase could be targeted by SaCas9/sgRNA during reactivation.

## Effect of HSV-1 ICP0 on editing

The HSV-1 ICP0 protein is known to reduce chromatin loading on the HSV-1 genome (*Cliffe and Knipe, 2008*; *Lee et al., 2016*), to promote the degradation of the catalytic subunit of DNA-dependent protein kinase, DNA-PKcs (*Lees-Miller et al., 1996*), and to inhibit DNA PK-dependent signaling (*Smith et al., 2014*) necessary for double stranded end joining. Therefore, ICP0 could impact CRISPR-Cas9 by at least two mechanisms. To evaluate the potential effect of ICP0 on editing of HSV-1 DNA, we performed qPCR experiments using ICP0-null HSV-1 *d*Prom (*Lee et al., 2016*). ICP0-null HSV-1 virus input DNA showed reduced levels of $U_L30$ (but not of UL29) PCR products in the presence of UL30-5 compared to the absence of UL30-5 (*Figure 9A B*). However, the difference was consistently smaller at all time points with ICP0-null HSV-1 compared to WT HSV-1 (*Figure 6B C*), suggesting that ICP0 contributes to the loss of viral sequences through effects on viral chromatin or effects on DNA damage repair.

## No detectable editing by UL30-5 sgRNA at predicted off-target sites within the human and HSV-1 genomes

To evaluate the specificity of our most efficient sgRNA (UL30-5) and determine whether our CRISPR/Cas9 gene targeting strategy against HSV induced any undesired off-target effects in the human genome, we further analyzed the whole genome sequencing data of HFFs expressing either SaCas9/UL30-5, SaCas9 only, or a mock plasmid and infected with WT HSV-1 (*Figure 7*). Comparing UL30-5 treated HFFs to the control with no Cas9, we identified 175,104 possible variants of which 25,300 were indels. Comparing UL30-5 treated HFFs to the control with no Cas9/no sgRNA, we called 172,468 possible variants of which 24,447 were indels. Of the 437 possible UL30-5 off-target sites identified through sequence analysis (Cas-OFFinder), none overlap with any of the variant calls (*Figure 7—source data 1*), suggesting that CRISPR/Cas9-induced editing at undesired sites did not occur. Next, we evaluated any potential off-site cleavage by UL30-5 within the HSV-1 genome itself. There were 87 possible HSV variants called in HSV-1 DNA, comparing the UL30-5 treated sample to the no Cas9/no sgRNA control, and 78 possible HSV variants to the no Cas9 control. Cas-OFFinder did not identify any predicted off-target sites with six or less mismatches within HSV-1 KOS DNA (GenBank: KT899744) except the predicted UL30-5 target site in $U_L30$. This argued that overt editing of the HSV-1 genome at loci other than the predicted UL30-5 target site did not occur at detectable levels.

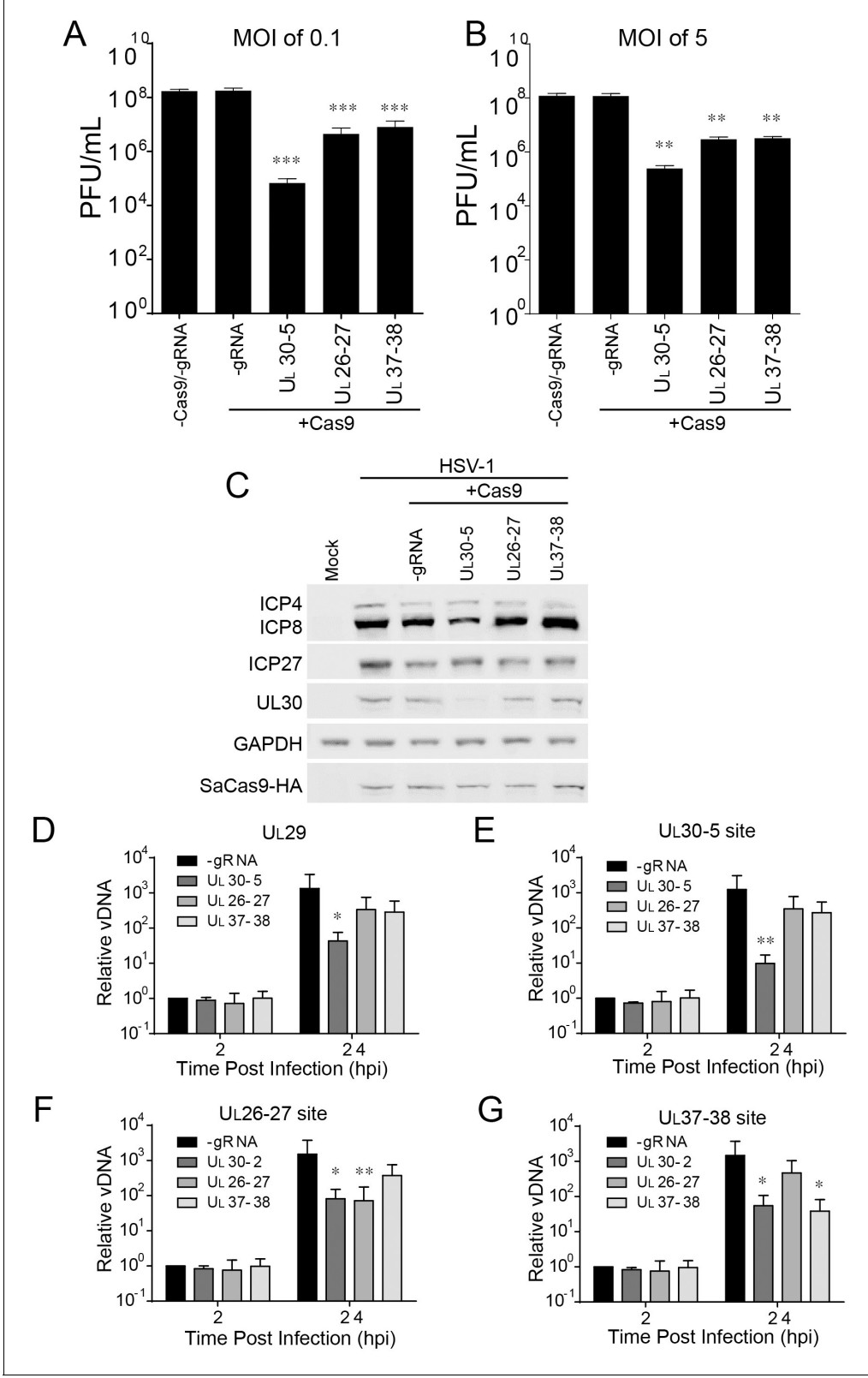

**Figure 8.** Effect of CRISPR-Cas9 on non-coding and non-essential regions of the HSV-1 genome. (**A and B**) HFFs transduced with lentivirus expressing SaCas9 and sgRNAs were infected with HSV-1 at an MOI of 0.1 (**A**) or 5 (**B**), harvested at 48 hpi or 24 hpi respectively. Viral yields were determined by plaque assays. The histogram shows the mean values and standard deviations from biological replicates (N = 4). (**C**) HFFs transduced with lentivirus expressing SaCas9 and sgRNA were infected with HSV-1 at an MOI of 5 and harvested at 10 hpi. Proteins were detected using immunoblotting with

*Figure 8 continued on next page*

*Figure 8 continued*

antibodies specific for the indicated proteins. GAPDH was shown as a control. (**D–G**) HFFs transduced with lentivirus expressing SaCas9 and sgRNA were infected with HSV-1 at an MOI of 3 and harvested at the indicated time post infection. The accumulated DNAs were detected by real time PCR amplifying in the $U_L29$ gene (ICP8, (**D**) or over the UL30-5 (**E**), UL26-27 (**F**), and UL37-38 (**G**) sgRNA target sites. The histogram shows the mean values and standard deviations from biological replicates (N = 4, (**A and B**) one-way ANOVA with Dunnett's multiple comparisons test, (**D–G**) Ratio paired t test, *p<0.05, **p<0.01, and ***p<0.001).

The online version of this article includes the following figure supplement(s) for figure 8:

**Figure supplement 1.** PCR amplification across the UL26-27 or UL37-38 target site in quiescent d109 genomes and reactivation.

Overall, our results are consistent with a model in which: (1) CRISPR/Cas9 can efficiently induce indel mutations in quiescent HSV-1 genomes; (2) replicating/replicated viral DNA is more susceptible to cleavage by CRISPR/Cas9 than quiescent HSV-1; (3) a fraction of viral DNA including input viral genomes is cleaved and not re-ligated during lytic replication; and (4) large deletions of viral DNA are induced during replication through DNA degradation. As a result, viral inhibition by CRISPR/Cas9 is mediated by differential mechanisms during lytic infection and latent infection, that is cleavage and non-rejoining resulting in large deletions, versus quiescent infection, with small indel formation (*Figure 10*).

## Discussion

There is a continuing medical need for antiviral strategies to target the latent DNA genomes of viruses such as HIV and the herpesviruses. While some have reported editing of HIV proviruses (*Ebina et al., 2013*), latent EBV genomes (*van Diemen et al., 2016*) and lytic HSV genomes (*Roehm et al., 2016*; *van Diemen et al., 2016*), there have been no reports of editing and inactivating latent HSV genomes. We performed in vitro screening of sgRNAs targeting the HSV-1 genome to identify sgRNAs capable of promoting efficient cleavage of viral DNA sequences. The most efficient gRNAs were then tested for their ability to edit HSV-1 viral genomes during lytic and quiescent infection. We observed robust editing of both types of viral genomes, but editing of the quiescent viral DNA was less efficient than editing of the lytic replicating genomes. The effects of the editing were also different in that we observed small indels in the edited quiescent genomes but extensive loss of viral sequences around the sgRNA target site and linear molecules in the edited lytic

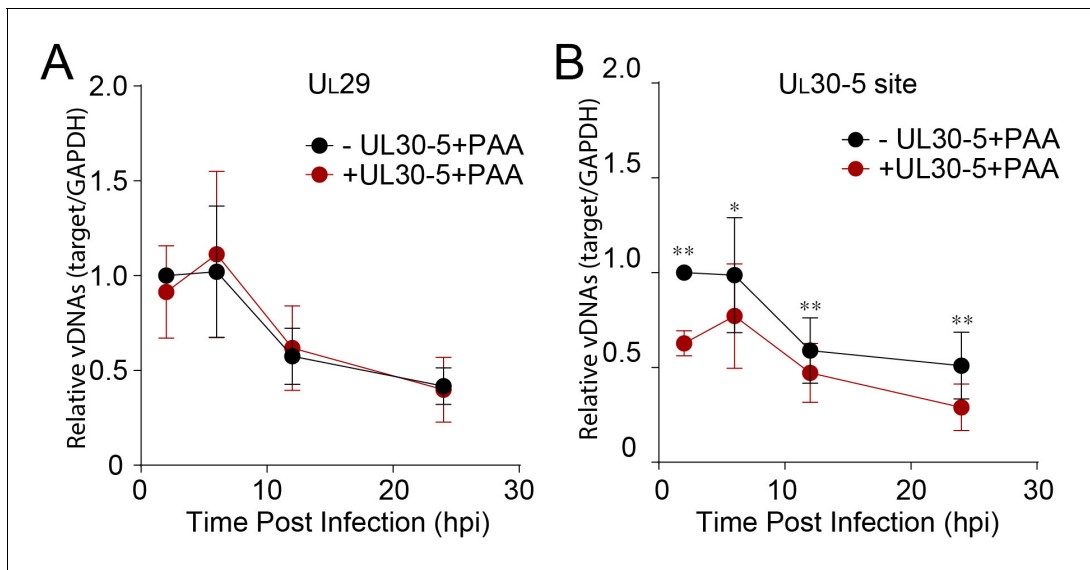

**Figure 9.** Effect of ICP0 on CRISPR-Cas9-induced DNA repair of input HSV genome. HFFs transduced with lentivirus expressing SaCas9 and sgRNA were infected with ICP0-null mutant HSV-1 at an MOI of 3 in the presence of PAA and harvested at the indicated time post infection. The accumulated DNAs were detected by real-time qPCR amplifying within the $U_L29$ gene (**A**) or over the UL30-5 sgRNA (**B**) target site. The histogram shows the mean values and standard deviations biological replicates (N ≥ 3, Ratio paired t test, *p<0.05 and **p<0.01).

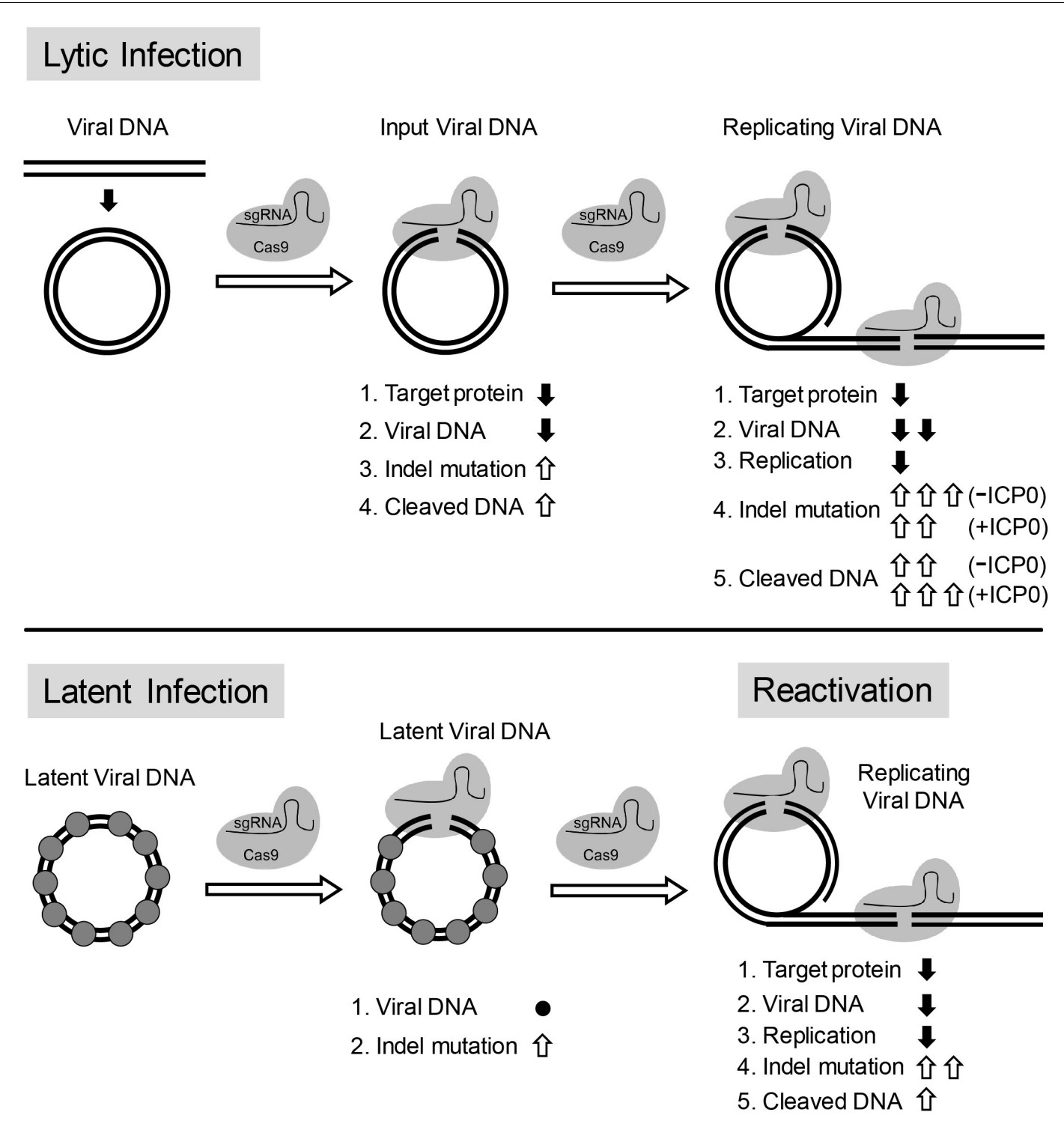

**Figure 10.** Model for CRISPR/Cas9 mediated inhibition of HSV lytic replication, editing of latent HSV genomes, and inhibition of reactivation of latent HSV. <u>Lytic infection</u>: Cas9/sgRNA cleaves input viral DNA. In the absence of viral DNA replication, either prior to the onset of viral replication or in the presence of PAA, the expression of Cas9/sgRNA targeting viral gene encoded protein is reduced and the input viral DNAs decrease. Cas9/sgRNA induces low levels of indel mutations at the sgRNA target site of the input viral DNA, and cleaved input viral DNA is accumulated. During viral DNA replication, expression of Cas9/sgRNA targeting the viral gene encoded protein is reduced, non-mutated template and its replicated viral DNAs are targeted by SaCas9/sgRNA, which results in a decrease of viral DNA, an increase in indel mutations and accumulation of cleaved viral DNA. Viral protein ICP0 contributes to Cas9/sgRNA-mediated editing/cleavage by removing histones and preventing DNA repair. <u>Quiescent infection</u>: Cas9/

*Figure 10 continued on next page*

*Figure 10 continued*

sgRNA induces indel mutations to viral DNA without significant change in latent viral DNA levels. (●: nucleosome) <u>Reactivation</u>: Cas9/sgRNA induces more indel mutations in non-mutated viral DNA and accumulation of cleaved viral DNA, which results in decrease of the expression of SaCas9/sgRNA targeted gene encoded protein, viral DNA, and viral replication. (⇧: increase, ⬇: decrease, •: no change).

genomes. These results show that the effects of CRISPR-Cas9 editing are different for lytic versus quiescent HSV genomes, likely due to the lytic viral functions such as ICP0 decreasing the chromatin loading of lytic genomes and inhibiting the host DNA damage responses. We further suggest a mechanistic model how CRISPR/Cas9 inactivates the circular HSV genome for lytic replication. We propose that cleaved genomes are either re-ligated by cellular repair mechanisms resulting in indel formation and disruption of viral gene expression or remain in a linear state during lytic infection, which is incompatible with mechanisms of viral DNA synthesis and subject to DNA degradation.

## Editing of quiescent viral DNA

We observed efficient editing of quiescent genomes with the UL30-5 sgRNA, with 40–80% of the viral genomes showing small indels. The numbers of quiescent genomes were not significantly affected by this editing, but reactivation was reduced, showing the inactivation of the quiescent viral genome. This editing is similar to that observed with editing of cellular genes, which is consistent with HSV-1 latent DNA being loaded with nucleosomes similar to cellular chromatin (*Cliffe et al., 2009*; *Deshmane and Fraser, 1989*; *Wang et al., 2005*). This is the first evidence, to our knowledge, of the ability of CRISPR/Cas9 to edit quiescent or latent HSV genomes efficiently.

The titers of reactivating virus were inhibited by as much as 1000-fold, much greater than the 50–80% editing of the quiescent viral genomes; thus, the sgRNA is likely targeting the reactivating genome as well as the quiescent viral genome. These results support the potential use of CRISPR-Cas9 to target reactivating virus as well as the latent virus as a therapeutic strategy.

## Editing of lytic viral DNA

We observed very efficient editing of the lytic replicating viral genomes, with reduced viral DNA replication, enhanced loss of sequences at the sgRNA target site, and generation of linear molecules. We hypothesize that SaCas9 cleaves the lytic viral DNA, but ICP0 or other viral proteins inhibit the re-joining of the viral DNA. As a result, linear molecules remain, and cellular exonucleases chew back the free DNA ends causing loss of sequences around the sgRNA target site (*Figure 10*).

The viral IE ICP0 protein is known to promote the removal of histones from viral input genomes (*Cliffe and Knipe, 2008*; *Lee et al., 2016*), at least in part by promoting the degradation or inactivation of host restriction factors such as PML, IFI16, Sp100, and ATRX, which promote epigenetic silencing of the HSV genome (*Cabral et al., 2018*; *Glass and Everett, 2013*; *Lukashchuk and Everett, 2010*; *Orzalli et al., 2013*; *Orzalli et al., 2012*). ICP0 is also known to inhibit DNA damage responses such as non-homologous end joining by promoting degradation of the catalytic subunit of the DNA-dependent protein kinase (*Lees-Miller et al., 1996*). We observed that ICP0 promotes the editing of HSV-1 input genomes in the absence of viral DNA replication, and this effect could be mediated by chromatin removal or inhibition of NHEJ or both.

Viral DNA replication reduces histone loading to HSV-1 DNA, by promoting either the removal or dilution of the associated histones (*Cliffe and Knipe, 2008*; *Oh and Fraser, 2008*) and makes the genome more accessible to RNA Pol II, TBP, and TAF1 (*Dremel and DeLuca, 2019*).

We found that viral DNA replication allows a novel editing effect on the viral target DNA (in addition to indel formation/mutagenesis) with cleavage resulting in extensive loss of genomic viral sequence at the sgRNA target site and lack of re-ligation leading to persistence of truncated linear viral DNA molecules both of which inhibit viral replication/function.

Thus, our studies point to several mechanisms by which CRISPR-Cas9 editing inhibits lytic infection: (1) Reduced levels of essential viral proteins due to Cas9 binding to essential genes and blocking their transcription, DNA cleavage reducing transcription of the essential genes, or frameshift mutations due to the indels introduced; and/or (2) Cleavage of viral DNA, chew-back by cellular nucleases and lack of re-ligation leading to linear viral DNA molecules that cannot participate in viral DNA synthesis by the rolling circle mechanism or a recombination-based mechanism. Therefore,

CRISPR-Cas9 editing can inhibit lytic infection at the levels of viral gene transcription, DNA replication, and protein expression.

## Comparisons with other HSV editing systems

It is unclear why our CRISPR-Cas9 system shows much higher rates of indel formation than those in previous reports (*Roehm et al., 2016*; *van Diemen et al., 2016*). However, our study differs from the previous report in several aspects, and all may contribute. First, we designed our sgRNAs to target the most proximal 5' sequence of open reading frames and tested sgRNAs targeting coding sequences (CDS) to disrupt the N-termini of viral proteins and directly screened for efficient indel formation in human cells using a model system for HSV-1 latency. Second, we used a single vector lentivirus system to introduce Cas9 and sgRNA followed by puromycin selection. The dual vector system used by others (*van Diemen et al., 2016*) may alter the ratio of Cas9 and sgRNA expression in individual cells and affect the efficiency of cleavage. Third, our study used the HSV-1 *d*109 mutant virus instead of VP16-eGFP mutant HSV-1 (HSV-1-eGFP). HSV-1-eGFP has all of the IE genes intact while *d*109 does not express any active IE genes. It is possible that HSV-1-eGFP might express low levels of ICP0 during quiescent infection, which may affect the efficiency of cellular repair of viral DNA. Fourth, we used *Staphylococcus aureus* Cas9 (SaCas9) instead of *Streptococcus pyogenes* Cas9 (SpCas9). It has been reported that SpCas9 and SaCas9 exhibit comparable editing efficiencies (*Friedland et al., 2015*); however, it is possible that SaCas9 may be more effective than SpCas9 in our assay. Because we designed our sgRNAs for both types of Cas9, we could test SpCas9 with our sgRNAs to evaluate differences in efficiency.

## Comparisons of different sgRNAs

At present, it remains unknown why UL30-5 is consistently more efficient than other sgRNAs, with several sgRNAs showing no or minor cleavage in cells despite efficient cleavage in our cell-free assay. We performed the cell-free assay using SpCas9 instead of SaCas9 because SaCas9 protein was not commercially available at the time. Differences in efficacy of individual sgRNAs in the in vitro cleavage assay and cell-based assays may therefore be due to sequence-dependent interactions between sgRNA and different Cas9 orthologs. In addition, the chromatin structure of targeted regions of viral DNA can affect CRISPR/Cas9 efficiency (*Horlbeck et al., 2016*), especially heterochromatin of latent genomes. Thus, the UL30-5 target site may be particularly accessible relative to other SaCas9/sgRNA complexes. This could point towards a more focused design strategy for sgRNAs targeting latent DNA viruses by generating an accessibility map of quiescent/latent genomes using ChIP-seq or ATAC-seq. Another explanation could be potential off-target effects of UL30-5 on other regions of the HSV-1 or the human genome. However, we did not identify off-target sequences recognized by UL30-5 in both genomes. Of note, knockout of UL30 also significantly reduced ICP8 (encoded by $U_L29$) expression, implying that targeting the 5' sequence of UL30 may interrupt expression from the nearby $U_L29$ promoter likely due to loss of viral genomic sequence. Alternatively, UL30 may be required for ICP8 expression and/or the stability of ICP8. However, UL30-2 showed an equivalent ICP8 reduction compared to UL30-5, implying that the UL30-5 sgRNA exerts additional effects on viral replication. Further studies are needed to understand the effect of UL30 on ICP8 and other viral proteins.

## Implications for the use of CRISPR/Cas9 as an antiviral strategy against HSV

We have established conditions where CRISPR/Cas9 can edit either quiescent or lytic viral genomes; thus, it could be used to target either latent genomes, lytic infection, or reactivating genomes. Further studies are in progress to determine if CRISPR/Cas9 can edit the HSV genome during latent infection in the resting sensory neuron host cell and in vivo models. While the potential for off-target mutations could make the risk too high for routine HSV prevention or therapy, this treatment might be appropriate in certain situations such as drug-resistant HSV infection in recurrent keratitis and herpes simplex encephalitis, which can lead to serious diseases. The novel form of efficient and extensive editing of the replicating HSV genome by CRISPR-Cas9 argues that this could be an important new way of controlling acute lytic infection or reactivating infection. For example, mucosal

delivery of CRISPR/Cas9 could be a useful treatment to prevent HSV infection or to reduce recurrent infections.

Even with the identification of sgRNAs that can target quiescent HSV-1 genomes, challenges remain in the application of this technology in vivo. Further studies are needed to understand how to deliver Cas9 and sgRNAs to latently infected sensory or other neurons in vivo. Editing in the resting neurons may also provide new challenges for gene editing. Sequence variability is potentially a problem, but we have found that the target site for the UL30-5 sgRNA is conserved in 45 HSV-1 genomes that we analyzed (*Szpara et al., 2014*, and other sequence available strains in https://www.ncbi.nlm.nih.gov/nuccore/?term=((herpes+simplex+virus+1+genome))).

In this study we found that HSV-1 lytic functions promote novel effects of viral DNA editing by CRISPR-Cas9 by reducing the chromatin loading of the viral genome and inhibiting host cell DNA repair. Thus, CRISPR/Cas9 may find the Achilles heel of HSV replication, the need to remove the silencing chromatin that the host cell puts on it in order to replicate and transcribe its genome, providing a new therapeutic approach for lytic infection and reactivation of herpes simplex virus.

# Materials and methods

**Key resources table**

| Reagent type (species) or resource | Designation | Source or reference | Identifiers | Additional information |
|---|---|---|---|---|
| Gene (*Staphylococcus aureus*) | SaCas9 | Addgene | pX601, Cat. #: #61591 | |
| Peptide, recombinant protein | SpCas9 | NEB | Cat. #: M0386 | |
| Cell line (*Homo-sapiens*) | HFF (Hs27) | ATCC | Cat# CRL-1634, RRID:CVCL_0335) | |
| Cell line (*Homo-sapiens*) | U2OS | ATCC | Cat# HTB-96, RRID:CVCL_0042 | |
| Cell line (*Chlorocebus sabaeus*) | Vero | ATCC | Cat# CCL-81, RRID:CVCL_0059 | |
| Antibody | Anti-ICP8 (Rabbit serum) | *Knipe et al., 1987* | | 1:5000 |
| Antibody | Anti-ICP4 (Mouse monoclonal, purified from hybridoma cell line 58S (ATCC HB8183)) | *Showalter et al., 1981* | | 1:2000 |
| Antibody | Anti-ICP27 (Mouse monoclonal) | Eastcoast Bio | Cat. #: P1119 | 1:5000 |
| Antibody | Anti-GAPDH ([6C5], Mouse monoclonal) | Abcam | Cat. #: ab8245 | 1:10000 |
| Recombinant DNA reagent | | Addgene | lentiCRISPRv2, Cat #: 52961 | Cloning vector |
| Recombinant DNA reagent | pX601-AAV-CMV-SaCas9-T2A-mCherry | This paper | | Template of SaCas9-T2A-mCherry for lentiSaCas9-mCherry-Puro |
| Recombinant DNA reagent | lentiSaCas9-mCherry-Puro | This paper | | Cloned SaCas9 gene into lentiCRISPRv2 |
| Software, algorithm | ICE analysis toolbox | https://ice.synthego.com | | |
| Software, algorithm | bcbio-nextgen | https://github.com/bcbio/bcbio-nextgen | v1.15 | |

*Continued on next page*

*Continued*

| Reagent type (species) or resource | Designation | Source or reference | Identifiers | Additional information |
|---|---|---|---|---|
| Software, algorithm | MuTect2 | https://www.ncbi.nlm.nih.gov/pubmed?term=20644199 | v2 | |
| Software, algorithm | bwa-mem | https://arxiv.org/abs/1303.3997 | v0.7.17 | |
| Software, algorithm | Cas-OFFinder | https://www.ncbi.nlm.nih.gov/pubmed/24463181 | v2.4 | |

## Cells and viruses

HFF (Hs27), U2OS (known as U-2 OS), and Vero cells were obtained from the American Type Culture Collection. FO6 (*Samaniego et al., 1997*) cells were provided by Neal Deluca, and V27 cells were described previously (*Rice et al., 1989*). Upon receipt of a cell line (Original Vial), it was expanded in a separate quarantine facility and low passaged stocks were made for experiments. All of the cell lines have been regularly tested for mycoplasma contamination and they were all free from the contamination. The morphology of the cells have been regularly examined in culture by microscope. FO6 cells were maintained with 500 µg/mL of G418 (Gibco) and 300 µg/mL of hygromycin B (Invitrogen). V27 cells were maintained with DMEM supplemented with 500 µg/mL of G418. U2OS and Vero cells were maintained in Dulbecco's modified Eagle medium (DMEM; Life Technologies and CORNING) supplemented with 5% (v/v) fetal bovine serum (FBS; Life Technologies) and 5% (v/v) bovine calf serum (BCS; Life Technologies) and, 2 mM L-glutamine in 5% $CO_2$. HFF and 293 T cells were maintained in DMEM supplemented with 10% (v/v) FBS. FO6 were maintained with 500 µg/mL of G418 and 300 µg/mL of hygromycin B. V27 cells were maintained with DMEM supplemented with 500 µg/mL of G418.

The HSV-1 KOS wild-type (WT) strain (*Colgrove et al., 2016*; *Schaffer et al., 1970*) used in this study was grown and titrated on Vero cells. HSV-1 *d*109 (*Samaniego et al., 1998*) used in this study was re-isolated on U2OS ICP4/27 (*Miyagawa et al., 2015*) using two round of plaque purifications. HSV-1 *d*109 was grown and titrated on U2OS ICP4/27 and FO6 cells. Infections were conducted with virus diluted in phosphate-buffered saline (PBS) containing 0.1% glucose (wt/vol), 0.1% BCS (v/v) for 1 hr with shaking at 37°C. The medium was changed to DMEM containing 1% BCS and incubated at 37°C.

CRISPR target sites design sgRNA target sequences with PAM sequences compatible with both SaCs9 and SpCas9 were designed using the crispr.mit.edu website (Zhang lab, MIT) within ~1 kbp downstream of the start codon for the selected genes ($U_L29$, $U_L30$, $U_L54$, a4). CRISPR sgRNA sequences were selected by highest score for specificity and the least off-targets within the human genome, as provided by the online CRISPR design tool. For each virus gene target, multiple independent CRISPR sgRNAs were selected and used for in vitro sgRNA transcription or cloned into pX601-AAV-CMV-SaCas9-T2A-mCherry and pLV-sgRNA-SaCas9-T2A-mCherry-P2A-Puro for cell-based assays.

## In vitro cleavage assay

To generate templates for sgRNA transcription, gene-specific oligonucleotides containing the T7 (5'-TAATACGACTCACTATA-3') promoter sequence, the 20 base target site without the PAM, and a complementary region were annealed to a constant oligonucleotide encoding the reverse-complement of the tracrRNA tail. Briefly, a 60 nt oligo (sgRNA primer), containing the T7 promoter (TAATACGACTCACTATAGGNNNNNNNNNNNNNNNNNNNNGTTTTAGAGCTAGAAATAGCAAG), the 20 nt of the specific SpCas9 sgRNA DNA binding sequence (with the first two nucleotides replaced by GG) and a constant 23 nt tail for annealing, was used in combination with a 82 nt reverse oligo (AAAAAAGCACCGACTCGGTGCCACTTTTTCAAGTTGATAACGGACTAGCCTTATTTTAACTTGCTATTTCTAGCTCTAAAAC) to add the sgRNA invariable 3' end (tail primer, *Table 1*). sgRNA and tail primers were annealed and a double stranded DNA (dsDNA) template was generated using fill in of ssDNA overhangs with AmpliTaq Gold 360 Master Mix (Applied Biosystems) at 72°C for 1.5 hr. The dsDNA was purified using the Promega Wizard Clean up kit, and approximately 150 ng of DNA were used as template for a T7 in vitro transcription (IVT) reaction (MAXIscript T7 Transcription Kit

from Invitrogen). In vitro transcribed sgRNAs were DNAse treated and purified using the Omega EZNA PF kit. The concentration of the purified sgRNA was measured using a NanoDrop 2000. After purification, the sgRNA was diluted to 200 ng/µL and stored at –80˚C. Cas9–sgRNA complexes were constituted before cleavage by incubating Cas9 nuclease, *S. pyogenes* (purchased from NEB) and the in vitro transcribed sgRNA for 10 min at 37˚C in reaction buffer. dsDNA substrates (500–1700 bp) for Cas9 cleavage containing the sgRNA target sites were generated by PCR from HSV-1 viral genomic DNA and purified using the Promega Wizard Clean up kit. In some cases, dsDNA for cleavage assays was obtained as g-blocks (IDT). Cleavage assays were conducted in a reaction volume of 10 µL with final SpCas9 concentrations of 300 nM, 100 nM, 33 nM, and 0 nM, 400 ng sgRNA, 200 ng dsDNA substrate in 1xCas9 nuclease reaction buffer (NEB) at 37˚C for 30 min followed by addition of reaction stop buffer (30% glycerol, 0.5 M EDTA, 2% SDS in $H_2O$) and incubation at 80˚C for 10 min. The cleaved dsDNA was analyzed using gel electrophoresis.

## Plasmid construction

pX601-AAV-CMV-SaCas9-T2A-mCherry plasmid cloning: pX601-mCherry was generated via Gibson assembly of the BamHI/EcoRI digested pX601 plasmid (addgene plasmid #61591) and a g-block (gb_pX601-mCherry, IDT) containing T2A-mCherry with 19 bp overhangs. Plasmid DNAs were purified using the endotoxin-free Midiprep kit (Qiagen) and Sanger sequenced.

gb_pX601-mCherry

CAGGCAAAAAAGAAAAAGGGATCCTACCCATACGATGTTCCAGATTACGCTTACCCATACGA TGTTCCAGATTACGCTGGCAGTGGAGAGGGCAGAGGAAGTCTGCTAACATGCGGTGACG TCGAGGAGAATCCTGGCCCAATGGTGAGCAAGGGCGAGGAGGATAACATGGCCATCATCAAG- GAGTTCATGCGCTTCAAGGTGCACATGGAGGGCTCCGTGAACGGCCACGAGTTCGAGA TCGAGGGCGAGGGCGAGGGCCGCCCCTACGAGGGCACCCAGACCGCCAAGCTGAAGGTGAC- CAAGGGTGGCCCCCTGCCCTTCGCCTGGGACATCCTGTCCCCTCAGTTCATGTACGGC TCCAAGGCCTACGTGAAGCACCCCGCCGACATCCCCGACTACTTGAAGCTGTCC TTCCCCGAGGGCTTCAAGTGGGAGCGCGTGATGAACTTCGAGGACGGCGGCGTGGTGACCG TGACCCAGGACTCCTCCCTGCAGGACGGCGAGTTCATCTACAAGGTGAAGCTGCGCGGCAC- CAACTTCCCCTCCGACGGCCCCGTAATGCAGAAGAAGACCATGGGCTGGGAGGCCTCC TCCGAGCGGATGTACCCCGAGGACGGCGCCCTGAAGGGCGAGATCAAGCAGAGGCTGAAGC TGAAGGACGGCGGCCACTACGACGCTGAGGTCAAGACCACCTACAAGGCCAAGAAGCCCG TGCAGCTGCCCGGCGCCTACAACGTCAACATCAAGTTGGACATCACCTCCCACAACGAGGAC TACACCATCGTGGAACAGTACGAACGCGCCGAGGGCCGCCACTCCACCGGCGGCATGGAC- GAGCTGTACAAGTAAGAATTCCTAGAGCTCGCTGAT

gb_UL30-5

GCTCGAGGGGGCCCGGCCTCTAGACTCGAGGTTAACCTGCAGCGTCCCTGTAGTCTTCAACA TTAACAACTTTAAGTCCAGCAATTTGAGTTAAGGGTGTTGCTCTCAATGATTTCATTAATGGTTCAA TTTTTAATTTCTTTTCTTGAGGGCCTATTTCCCATGATTCCTTCATATTTGCATATACGATACAAGGC TGTTAGAGAGATAATTAGAATTAATTTGACTGTAAACACAAAGATATTAGTACAAAATACGTGACG TAGAAAGTAATAATTTCTTGGGTAGTTTGCAGTTTTAAAATTATGTTTTAAAATGGACTATCATATGC TTACCGTAACTTGAAAGTATTTCGATTTCTTGGCTTTATATATCTTGTGGAAAGGACGAAACACC- GACACGTGAAAGACGGTGACGGGTTTTAGAGCTAGAAATAGCAAGTTAAAATAAGGCTAGTCCG TTATCAACTTGAAAAAGTGGCACCGAGTCGGTGCTTTTTTATCCTGTACAGAATTCGAGCGCTTC TGCAAGGGCGAATTCTGGGTGCAAAGATGGATAAAG

lentiSaCas9-mCherry-Puro cloning: We cloned SaCas9, its trans-activating crRNA (tracrRNA), and mCherry into lentiCRISPRv2 plasmid. The lentiCRISPRv2 (addgene plasmid #52961) were digested using BsmBI (NEB) and BamHI (NEB) and purified using DNA purification kit (Zymo). DNA assembly was performed using NEBuilder HiFi DNA Assembly (NEB) and three DNA fragments, the purified linear lentiCRISPRv2, a double-stranded DNA gBlock (IDT) containing sgRNA cloning sites and tracrRNA sequences (IDT, 5' CTTTATATATCTTGTGGAAAGGACGAAACACCGGAGACGtGATAT-CaCGTCTCAGTTTTAGTACTCTGGAAACAGAATCTACTAAAACAAGGCAAAATGCCGTGTTTATC TCGTCAACTTGTTGGCGAGATTTTGAATTCGTAGACTCGAGGCGTTG ACATTG 3'), and PCR fragment containing SaCas9-mCherry according to the manufacturer's protocol. The SaCas9-mCherry was generated using primers (5' TGAATTCGTAGACTCGAGGCGTTG ACATTG 3' and 5' TCAGCA-GAGAGAAGTTTGTTGCGCCGGAACCGCTAGCCTTGTACAGCTCGTCCATGC 3') and pX601-AAV-CMV-SaCas9-T2A-mCherry as a template.

## Establishment of quiescent *d*109 infections in HFF cells

We established quiescent *d*109 HSV-1 infections in HFFs as described (*Ferenczy and DeLuca, 2011*) with modifications. Briefly, HFF cells were infected with *d*109 HSV-1 at an MOI of 10 in PBS containing 0.1% glucose (wt/vol) and 0.1% BCS (v/v) for 1 hr with shaking at 37°C and incubated at 37°C for 7–10 d in DMEM supplemented with 10% (v/v) FBS.

## Lentivirus preparation and transduction

Human 293T cells ($3 \times 10^6$) were plated in a 100 mm dish at 20–24 hr before transfection. The cells were transfected with lentiSaCas9-mCherry-Puro (5 µg), psPAX2 (4 µg), and pVSV-G (1 µg) using Effectene (Qiagen) according to the manufacturer's protocol or using polyethyleneimine (PEI, Polysciences, Inc #23966). For PEI transfection, a total of 10 µg of DNAs in 500 µL of Opti-MEMI (Gibco, #31985) and 30 µg of PEI in 500 µL of Opti-MEMI were mixed and incubated at ambient temperature for 20 min. The mixtures were added directly to 293T cells containing fresh 5 mL of DMEM (supplemented with 10% (vol/vol) FBS and 2 mM glutamine). The cells were incubated at 37°C for 8–12 hr, replaced with 10 mL of fresh DMEM supplemented with 10% (vol/vol) FBS and 2 mM glutamine, and incubated at 37°C. Then, at every 12–24 hr for 48–60 hr, the media were harvested and replaced with fresh DMEM 10%, and the harvested media were saved on ice. The collected media were pooled and filtered using 0.45 µm syringe filter (Pall) and kept on ice until lentivirus transduction. To transduce HFF cells with lentivirus for lytic infection assay, HFF cells were plated at a density of $2 \times 10^5$/well in a T25 flask one day prior to transduction and transduced with 2 mL of lentivirus containing 3–4 µg/mL of polybrene (Santa Cruz, sc-134220). To transduce quiescent *d*109 infected cells, the cells were transduced with 2 or 10 mL of lentiviruses containing 3–4 µg/mL of polybrene in 6-well plates or in T150 flasks. The next day, the transduction medium was replaced with fresh medium, and the cells were incubated for 2 d at 37°C followed by puromycin treatment (1 µg/mL) for 7–10 d.

## SDS-PAGE and immunoblotting

Immunoblotting was performed as described previously (*Oh et al., 2014*). Briefly, HFF cells were lysed in 1xNuPAGE sample buffer (Life Technologies) (*Oh et al., 2014*). The proteins were resolved in NuPAGE 4–12% Bis-Tris Gels (Life Technologies) and then transferred to a Nitrocellulose Membrane (Bio-Rad, #1620112). The membranes were blocked in Odyssey Blocking Buffer (LI-COR) then incubated with antibodies specific for HSV-1 ICP8 (1:5000, rabbit serum 3–83 [*Knipe et al., 1987*], HSV-1 ICP4 (1:2000, monoclonal mouse (mAb) 58S [*Showalter et al., 1981*], purified from hybridoma cell line 58S (ATCC HB8183)), ICP27 (1:5000, mAb, Eastcoast Bio), or GAPDH (1:10000, mAb, Abcam). The membranes were incubated with secondary antibodies, IRDye 680RD or IRDye 800 (LI-COR), for 45 min. Near-infrared fluorescence was detected using Odyssey (LI-COR). Protein expression level was quantified using Image J or ImageStudio V4 (LI-COR).

## Quantitative PCR

DNA was purified using the DNeasy Blood and Tissue kit (Qiagen) according to the manufacturer's protocol. Relative amounts of specific cDNAs were quantified using their specific primers with the SYBR Green PCR Master Mix reagent (Applied Biosystems) and the StepOnePlus Real-Time PCR system (Life Technologies). Primers (IDT) used in this study are as follows: *GAPDH* (5'-TTCGACAG TCAGCCGCATCTTCTT-3' and 5'-CAGGCGCCCAATACGACCAAATC-3' ; *Cliffe and Knipe, 2008*), *ICP8* (5'-GTCGTTACCGAGGGCTTCAA-3' and 5'-GTTACCTTGTCCGAGCCTCC-3'), *UL30* (5'-AAGACGTTCACCAAGCTGCT-3' and 5'-CAGATCCACGCCCTTGATGA-3'), UL30-5 (5'-CCCAAGG TGTACTGCGGG-3' and 5'-CTCCACGTTCTCCAGGATGT-3'), UL26-27 (5'-GTAGGCGGGGTAGC TTTACAAT-3' and 5'-GGCCTGTTTCCTCTTTCCTT-3'), and UL37-38 (5'-TGACTGTCGTGCGCTGTA-3' and 5'-CGGGATGCCGGGACTTA-3').

## LM-PCR

Ligation-mediated PCR (LM-PCR) was performed as described previously with modification (*Brinkman et al., 2018*). HFFs transduced with lentivirus expressing SaCas9 and sgRNA were infected with HSV-1 at an MOI of 3 and harvested at the indicated time post infection. Total DNA was purified, and a primer extension reaction was performed using a primer (0.1 µM) complementing downstream of UL30-5 site (UL30 63610 R: CAGAAGTTGTCGCACAGGTA) to convert all the

cleaved DNA (500 ng total) into blunt ended dsDNA toward to UL30-5 target site using Q5 2x master mix (NEB, final 50 µL, 35 s at 98℃, 15 s at 50℃, and 30 s at 72℃). The blunt end dsDNA was purified using PCRClean DX beads (Aline Biosciences) at 1:1.2 ratio following manufacturer's protocol. Modified NEB adaptor primers (NEB adaptor top: GATCGGAAGAGCACACGT and NEB blunt adaptor bottom: GACTGGAGTTCAGACGTGTGCTCTTCCGATC) were incubated at 95℃ for 5 min and annealed over 20 min by decreasing temperature gradually to 25℃. The annealed adaptor (50 nM) was ligated to the blunt end of dsDNA (30 µL out of 35 µL elution) with T4 ligase (NEB) in 50 µL reaction at 16℃ overnight. PCR was performed using a primer complementing the adaptor (NEB PCR primer: GCGACGTGTGCTCTTCCGATC) and UL30 63610 R primer using Q5 2x mater mix following manufacturer's protocol. Control PCR products were generated using a primer complementary to a site near the UL30-5 target site (UL30-5 cut 63181 F: TCACCGTCT TTCACGTGTA) and UL30 63610 R primer.

## Deep sequencing and indel analysis

sgRNA target sites were amplified using the following gene-specific primers with added adapter sequences: UL30-5_for 5'-CCCAAGGTGTACTGCGGG-3', UL30-5_rev 5'-CTCCACGTTCTCCAGGA TGT-3', UL29-3_for 5'-ATAGACTCGAGGGCCAGGG-3', UL29-3_rev 5'-TGACGAAAACCAC-GAGGGC-3', adapter_for 5'-TCTTTCCCTACACGACGCTCTTCCGATCT-3', adapter_rev 5'-TGGAG TTCAGACGTGTGCTCTTCCGATCT-3'. Amplified DNA was isolated from 2% agarose gels (Promega Wizard Clean up kit), barcoded and prepared for IlluminaMiSeq sequencing using the 16SMetagenomic Sequencing Library Preparation kit (Illumina). DNA concentrations were determined by the Qubit fluorometer 2.0 (Life Technologies, USA) with the Qubit dsDNA High Specificity assay kit. DNA libraries were sequenced by IlluminaMiSeq (150 bp paired-end).as previously described (*Gagnon et al., 2014*). Indel frequencies were determined by quantifying aligned reads containing insertions or deletions 1 bp or larger (http://www.outknocker.org/). Analysis of indel size was performed using the ICE analysis toolbox (https://www.synthego.com/products/bioinformatics/crispr-analysis).

## Whole genome sequencing

Cell pellets of approximately 1–5 million cells were digested overnight at 50℃ in 500 µL lysis buffer containing 100 µg ml−one proteinase K (Roche), 10 mM Tris (pH 8.0), 200 mM NaCl, 5% w/v SDS, 10 mM EDTA, followed by phenol:chloroform precipitation, ethanol washes, and resuspension in 10 mM Tris buffer (pH 8.0). Genomic DNA was then transferred to the Genomics Platform at the Broad Institute of MIT and Harvard for Illumina Nextera library preparation, quality control, and sequencing on the Illumina HiSeq X10 platform. Sequencing reads (150 bp, paired-end) were aligned to the GRCh38/hg38 reference genome using the BWA alignment program. Sequencing quality and coverage were analyzed using Picard tool metrics.

## gRNA off-target analysis

To study the frequency of off-target editing due to the UL30-5 gRNA, we screened the entire human and HSV-1 genome for indel formation at all predicted UL30-5 off-target sites with six or less mismatches. We transduced HFFs with SaCas9/UL30-5, UL30-5 only and a vector without SaCas9 and sgRNA as above, isolated genomic DNA and performed whole genome sequencing. To identify possible variants within the human genome in the UL30-5 treated HFFs relative to controls, we mapped WGS reads to the human genome (GRCh38/hg38) and used mutect2's tumor-normal calling treating the CRISPRed sample as a 'tumor'. We then used Cas-OFFinder (http://www.rgenome.net/cas-offinder/) in offline mode to look for all predicted off-target sites for SaCas9/UL30-5 with six or fewer mismatches within GRCh38/hg38 and determined the overlap between predicted variants and off-target sites. Next, we took the unmapped reads, aligned them to the HSV-1 KOS genomic sequence and called variants, again using mutect2's tumor-normal calling. We then determined the overlap between predicted variants and off-target sites within HSV-1 KOS.

## Statistical analysis

Biological replicates (as indicated in the legends, N) are independent experiments that are performed using the same test at different times. Technical replicates are the samples collected at the

same times at the same conditions multiple times. Biological replicates are analyzed to determine statistical significance. Statistical analysis was performed using Prism 6 (Version 6.01) software (GraphPad Software). Student's t-test, ratio paired t test, or one-way ANOVA with Dunnett's multiple comparison test were used to determine statistically significance with two-sided (95% confidence level). The histograms are mean values with standard deviations.

## Acknowledgements

This research was supported by NIH grant AI135423 (KE and DMK) and AI098681 (DMK), and a Q-FASTR Award from Harvard Medical University (DMK). We thank Jeho Shin for technical assistance and Patrick T Waters for preparation of the manuscript.

## Additional information

### Competing interests

David M Knipe: Reviewing editor, *eLife*. Hyung Suk Oh, Werner M Neuhausser, Magdalena Angelova, Kevin C Eggan: has patent applications pending. U.S. Patent application No. 62/365,826, International Patent Application PCT/US2017/043225. The other authors declare that no competing interests exist.

### Funding

| Funder | Grant reference number | Author |
| --- | --- | --- |
| National Institutes of Health | P01 AI098681 | David M Knipe |
| National Institutes of Health | R21 AI135423 | Kevin C Eggan |
| Harvard Medical School | Q-FASTR Award | David M Knipe |

The funders had no role in study design, data collection and interpretation, or the decision to submit the work for publication.

### Author contributions

Hyung Suk Oh, Conceptualization, Formal analysis, Validation, Investigation, Visualization, Methodology, Project administration; Werner M Neuhausser, Conceptualization, Formal analysis, Funding acquisition, Validation, Investigation, Visualization, Methodology, Project administration; Pierce Eggan, Investigation, Methodology; Magdalena Angelova, Conceptualization, Formal analysis, Validation, Investigation, Visualization, Methodology; Rory Kirchner, Formal analysis, Sequencing data analysis; Kevin C Eggan, Conceptualization, Methodology, Project administration; David M Knipe, Conceptualization, Supervision, Funding acquisition, Methodology, Project administration

### Author ORCIDs

Hyung Suk Oh (iD) https://orcid.org/0000-0002-1739-0389
Werner M Neuhausser (iD) https://orcid.org/0000-0002-5092-2658
David M Knipe (iD) https://orcid.org/0000-0003-1554-6236

### Decision letter and Author response

Decision letter https://doi.org/10.7554/eLife.51662.sa1
Author response https://doi.org/10.7554/eLife.51662.sa2

## Additional files

### Supplementary files

- Transparent reporting form

## Data availability

All data generated or analysed during this study are included in the manuscript and supporting files.

The following previously published datasets were used:

| Author(s) | Year | Dataset title | Dataset URL | Database and Identifier |
|---|---|---|---|---|
| Colgrove RC, Liu X, Griffiths A, Raja P, Deluca NA, Newman RM, Coen DM, Knipe DM | 2016 | Human herpesvirus 1 isolate KOS, complete genome | https://www.ncbi.nlm.nih.gov/nuccore/KT899744 | NCBI GenBank, KT899744.1 |
| Miga KH, Newton Y, Jain M, Altemose N, Willard HF, Kent WJ | 2014 | hg38 | https://genome.ucsc.edu/cgi-bin/hgGateway?db=hg38 | Genome Reference Consortium, Human GRCh38.p12 (GCA_000001405.27) |

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
