## [Decision Letter]

Thank you for submitting your work entitled "Herpesviral lytic gene functions render the viral genome susceptible to novel editing by CRISPR/Cas9" for consideration by *eLife*. Your article has been reviewed by three reviewers, including Sara Sawyer as the Reviewing Editor, and the evaluation has been overseen by Karla Kirkegaard as the Senior Editor.

As you can see, the reviews are overall positive about the paper, and we only ask that you address the minor revisions requested by reviewer 3.

Reviewer #1:

CRISPR-based treatments for serious viral infections are important to explore and develop. Here, the authors have made an advance in this area, particularly by targeting latent HSV-1 genomes. I have no major concerns.

Reviewer #2:

I thought this an interesting and technically sound study that examines in detail how Sa Cas9 can edit either latent or lytic HSV-1 genomes in cells in culture. There are a number of interesting, unexpected findings including the fact that editing of HSV-1 DNA in these different stages of the viral life cycle has quite different effects, with lytic DNA being both more susceptible and more damaged than quiescent genomes. As there is a lot of interest in using gene editing as a treatment approach to the cure of latent DNA virus infections, especially HSV-1 and -2, I think this paper will be of broad interest to both virologists and gene editing afficionados. I actually have no recommendations for improvement of this well written article and recommend acceptance as is.

Reviewer #3:

In the submitted manuscript, Oh and colleagues describe a detailed analyses of a CRISPR/Cas9 system to target HSV-1 genomes. Although most of the experiments evaluate lytic replication, the effects of selected reagents are also tested in quiescent infections. The authors analyze a number of sgRNAs, starting from cleavage in vitro. sgRNAs targeting coding and non-coding regions of the HSV-1 genome resulted in cleavage, indels, and large deletions (spanning some 50 kb). Replicating genomes were the most susceptible, and quiescent genomes the least.

The work is comprehensive and detailed and most of the conclusions are well supported by the data. Presentation of the data is very clear, with one exception.

The Discussion appears rather optimistic in that this approach will move toward clinical development relatively easily. There appear to be still many major hurdles on the path, including the delivery of the vector to the already latently infected neurons, the efficiency the editions, and sequence variability between clinical isolates.

Major points:

It would be prudent with the limited evidence available at this time to tone down speculations about possible clinical use, and include a brief discussion of the hurdles that still need to be overcome.

The detection of only recombinant viruses in the reactivation experiments (discussion of Figure 3 in the subsection “CRISPR/Cas9 inhibits reactivation of quiescent HSV-1 genomes”) is interpreted as indicating lack of reactivation. According to most of the data in the manuscript, wouldn't it just as likely that the quiescent genomes actually reactivated and then recombined and were cleaved while replicating?

Figure 7 is difficult to read due to the inherent noise of analyzing base by base. It would likely be much easier to read if the reads were presented as ratio (treatment/control) at each position. The ratio for 2/3 of the genomes would be close to 1, and for the remaining 1/3 would be around a quarter. This approach would eliminate the visual interference of the inherent noise in the number of reads analyzed by base pair. Panel B is incompletely described (it appears that this panel presents the reads using a moving window).

"Full cleavage" and "partial cleavage" (legend to Table 1) are not defined.

If the DNA samples were still available, the percentage of cleaved genomes in Figure 5F/G could be easily quantitated with a PCR across the cleavage site and one at each side of it. This would provide useful information.

Figure 7—source data 1 appears not to have any labels or description.

---

## [Author Response]

Reviewer #3:In the submitted manuscript, Oh and colleagues describe a detailed analyses of a CRISPR/Cas9 system to target HSV-1 genomes. Although most of the experiments evaluate lytic replication, the effects of selected reagents are also tested in quiescent infections. The authors analyze a number of sgRNAs, starting from cleavage in vitro. sgRNAs targeting coding and non-coding regions of the HSV-1 genome resulted in cleavage, indels, and large deletions (spanning some 50 kb). Replicating genomes were the most susceptible, and quiescent genomes the least.The work is comprehensive and detailed and most of the conclusions are well supported by the data. Presentation of the data is very clear, with one exception.The Discussion appears rather optimistic in that this approach will move toward clinical development relatively easily. There appear to be still many major hurdles on the path, including the delivery of the vector to the already latently infected neurons, the efficiency the editions, and sequence variability between clinical isolates.Major points:It would be prudent with the limited evidence available at this time to tone down speculations about possible clinical use, and include a brief discussion of the hurdles that still need to be overcome.

Interestingly, another reader felt that we were very pessimistic about the application of CRISPR therapy to HSV infection in this text. In any event, we have revised the text in the last section of the Discussion to better describe the remaining challenges in delivery to and editing in sensory neurons, and we have stated that the UL30-5 gRNA target site is conserved in the genomes of 45 strains and isolates.

The detection of only recombinant viruses in the reactivation experiments (discussion of Figure 3 in the subsection “CRISPR/Cas9 inhibits reactivation of quiescent HSV-1 genomes”) is interpreted as indicating lack of reactivation. According to most of the data in the manuscript, wouldn't it just as likely that the quiescent genomes actually reactivated and then recombined and were cleaved while replicating?

We actually said that the inhibition of reactivation was “near complete”, not that there was a lack of reactivation. We do agree that the recombinants are likely formed following reactivation. To clarify this section, we have revised the text to focus on the very low levels of reactivation, not the fact that they are recombinants.

Figure 7 is difficult to read due to the inherent noise of analyzing base by base. It would likely be much easier to read if the reads were presented as ratio (treatment/control) at each position. The ratio for 2/3 of the genomes would be close to 1, and for the remaining 1/3 would be around a quarter. This approach would eliminate the visual interference of the inherent noise in the number of reads analyzed by base pair. Panel B is incompletely described (it appears that this panel presents the reads using a moving window).

We are not sure what normalization the reviewer is requesting because normalizing to treatment/control does not give a ratio of 1 due to reduced viral DNA amplification in the presence of the sgRNAs. When we plotted the data as normalized to the control cell, the levels of viral DNA in the edited sample were low and the loss of specific

sequences was not so clear. Because the goal of Figure 7 was to show the loss of sequences near the sgRNA target site, we have chosen to keep the figure as before.

"Full cleavage" and "partial cleavage" (legend to Table 1) are not defined.

We thank the reviewer for catching this and have added descriptions of “Full cleavage” and “partial cleavage” in a legend for Table 1.

If the DNA samples were still available, the percentage of cleaved genomes in Figure 5F/G could be easily quantitated with a PCR across the cleavage site and one at each side of it. This would provide useful information.

We did perform PCR across the target site and another site in Figure 6B-D on another set of samples.

Figure 7—source data 1 appears not to have any labels or description.

We thank the reviewer for catching this and have added legends for Figure 7—source data 1.